# Inhibition of *Clostridium difficile* TcdA and TcdB toxins with transition state analogues

Ashleigh S. Paparella [1], Briana L. Aboulache [1,4], Rajesh K. Harijan[1], Kathryn S. Potts [2], Peter C. Tyler [3] & Vern L. Schramm [1✉]

*Clostridium difficile* causes life-threatening diarrhea and is the leading cause of healthcare-associated bacterial infections in the United States. TcdA and TcdB bacterial toxins are primary determinants of disease pathogenesis and are attractive therapeutic targets. TcdA and TcdB contain domains that use UDP-glucose to glucosylate and inactivate host Rho GTPases, resulting in cytoskeletal changes causing cell rounding and loss of intestinal integrity. Transition state analysis revealed glucocationic character for the TcdA and TcdB transition states. We identified transition state analogue inhibitors and characterized them by kinetic, thermodynamic and structural analysis. Iminosugars, isofagomine and noeuromycin mimic the transition state and inhibit both TcdA and TcdB by forming ternary complexes with Tcd and UDP, a product of the TcdA- and TcdB-catalyzed reactions. Both iminosugars prevent TcdA- and TcdB-induced cytotoxicity in cultured mammalian cells by preventing glucosylation of Rho GTPases. Iminosugar transition state analogues of the Tcd toxins show potential as therapeutics for *C. difficile* pathology.

[1] Department of Biochemistry, Albert Einstein College of Medicine, Bronx, NY, USA. [2] Department of Developmental and Molecular Biology and Gottesman Institute of Stem Cell Biology and Regenerative Medicine, Albert Einstein College of Medicine, Bronx, NY, USA. [3] The Ferrier Research Institute, Victoria University of Wellington, Lower Hutt, New Zealand. [4] Present address: Department of Biochemistry, University of Colorado Boulder, Boulder, CO, USA. ✉email: vern.schramm@einsteinmed.org

Clostridium difficile is a Gram-positive spore forming bacterium and is the leading cause of healthcare associated infections in the United States. The resulting annual excess healthcare costs are estimated to be as high as $4.8 billion[1,2]. C. difficile infections (CDI) typically arise following treatment with broad-spectrum antibiotics that disrupt the normal gut microbiota, allowing C. difficile spores to proliferate and to populate the gut. Toxin release from C. difficile cells leads to an inflammatory infection of the colon and potentially fatal diarrhea[3,4]. Current treatment options include removing the antibiotic responsible in order to encourage restoration of the normal gut microbiota or to administer additional antibiotics such as vancomycin or fidaxomicin to eradicate the pathogen[5,6]. However, additional antibiotic therapy prevents re-establishment of the healthy gut microbiota, leading to recurrences of CDI in approximately 20% of patients[7]. Furthermore, C. difficile has acquired resistance to many antibiotics, highlighting the urgency for new treatments[8]. Our strategy to treat CDI is to develop enzymatic transition state analogues to target the virulence factors of C. difficile, TcdA and TcdB, the primary determinants of disease pathogenesis[9]. A third virulence factor, CDT binary toxin is produced by a minority of clinical isolates of C. difficile and can contribute to disease when present[10,11].

TcdA and TcdB are 308 kDa and 270 kDa, respectively, multi-domain protein toxins sharing 49% sequence identity[12]. The mechanism of action of TcdA and TcdB has been reviewed extensively[12–14]. TcdA and TcdB bind to target cells via a C-terminal receptor binding domain, triggering internalization by clathrin-mediated endocytosis. Acidification of endosomes causes pH-dependent conformational changes, leading to the formation of a trans-membrane pore and subsequent delivery of the N-terminal auto-processing cysteine protease domain (CPD) and glucosyltransferase domains (GTD) into the cytosol. The CPD is allosterically activated by intracellular inositol hexakisphosphate, catalyzing the release of the GTD into the cytosol[12–14]. There, GTD glucosylates and inactivates Rho GTPases, including Rac1 and Cdc42 at Thr35 and RhoA at Thr37 in the switch I effector region using UDP-glucose as the glucosyl donor[12–16]. Inactivation of Rho GTPases by the Tcd toxins causes actin-depolymerization resulting in a loss of structural integrity of the cell and eventually cell death through caspase-3 and caspase-9 dependent pathways[13,17] (Fig. 1a). Although both toxins exhibit the same mechanism of cellular toxicity, it has been shown that TcdB is more potent than TcdA in its ability to induce cell rounding and cell death[18,19]. Multiple studies to investigate the virulence of TcdA+ TcdB-, TcdA- TcdB+ and TcdA+ TcdB+ strains of C. difficile demonstrated that a TcdA+ TcdB- strain was less virulent than both TcdA- TcdB+ and TcdA+ TcdB+ strains[20,21]. Studies performed by Kuehne et al[9] in a hamster model demonstrated that a TcdA+ TcdB- strain was almost as virulent as TcdA+ TcdB+ and TcdA- TcdB+ strains. Despite differences in species sensitivity, the sum of these studies indicates that both TcdA and TcdB are major determinants of C. difficile pathogenesis.

Therapeutic targeting of TcdA and TcdB is an emerging strategy to treat CDI. By targeting the Tcd toxins directly, damage to the human colon is minimized and importantly, the human gut microbiota is spared. A healthy gut microbiome is a critical barrier in preventing CDI[7]. Recent developments to target the Tcd toxins include Bezlotoxumab (Zinplava™), a monoclonal antibody that acts by recognizing two epitopes in the C-terminal receptor binding domain of TcdB[22]. Other recent studies report the development of small molecule inhibitors targeting the CPD[23] and the GTD domains[24,25]. Here, we propose inhibition of the GTD domains of both Tcd toxins with transition state analogues of the Tcd catalyzed reaction.

Knowledge of the transition state structure of an enzymatic reaction provides a blueprint for the design of transition state analogues[26]. By mimicking the geometry and electrostatic potential of an enzymatic transition state, transition state analogues bind tightly to their cognate enzymes to provide an attractive class of small molecule inhibitors[26]. The GTD domains of TcdA and TcdB catalyze transfer of a glucosyl residue to a nucleophilic threonine acceptor in Rho GTPases. In addition to glucosyltransferase (GT) activity, TcdA and TcdB also catalyze the hydrolysis of UDP-glucose in the absence of Rho GTPases[27] (Fig. 1b). Although there is no known physiological significance, TcdA and TcdB GTD domains exhibit glucohydrolase (GH) activity at a significantly slower rate than glucosyl transfer to Rho GTPases[27,28]. GT and GH enzymes catalyze the transfer of glucose to an acceptor with either inversion or retention of stereochemistry at the anomeric carbon[29]. All GT and GH catalyzed reactions that have been well-characterized involve the formation of a glucocation-like transition state where the anomeric carbon is near-sp$^2$ hybridized and positive charge develops around the anomeric carbon[29,30].

Here we use kinetic isotope effects (KIEs) to determine the transition state features of the TcdA and TcdB GH reactions. Both enzymes demonstrate considerable glucocation character. Iminosugars that have similarity to the glucocationic transition state were tested for inhibitory activity against the TcdB and TcdA GTD domains (TcdB-GTD and TcdA-GTD). Isofagomine and noeuromycin inhibit both TcdA and TcdB by binding in the catalytic sites with structures consistent with glucocationic transition state analogues. Both inhibitors prevent Tcd-induced toxicity of cultured mammalian cells. The transition state analogue inhibitors serve as candidates with the potential to be developed as new treatment options for CDI.

## Results

**KIE analysis of TcdA and TcdB glucohydrolase reaction.** Enzymatic transition state structures can be probed with kinetic isotope effects (KIEs). KIEs compare the chemical steps of enzymatic reaction rates with isotopically labeled and unlabeled substrates and can be used to measure the degree of glucocation character for GH and GT transition states[29,30]. Isotopically labeled molecules differ from their natural abundance counterparts by having an altered zero point energy (ZPE), the sum of the energies of its bond vibrations in its ground state. The altered ZPE requires isotopically labeled molecules to experience different energy barriers to reach the transition state and this difference forms the basis of KIEs. If a specific atom is weakly bonded at the transition state, heavy isotopic substitution will result in a slower rate, exhibiting a normal KIE ($k_{light}/k_{heavy} > 1$). Atoms more strongly bonded at the transition state, exhibit inverse KIEs ($k_{light}/k_{heavy} < 1$)[31]. KIE measurements are determined by placing heavy isotope atoms at every position in a substrate molecule expected to be perturbed at the transition state. The KIEs provide a description of the geometry and electron distribution at the transition state. KIEs were measured by the competitive radiolabeled approach with isotopically labeled UDP-glucose substrates. This method yields (V/K) KIEs which report on all steps from substrate binding up to and including the first irreversible chemical step[32,33]. The slow hydrolysis of UDP-glucose by TcdA and TcdB were used to determine the nature of the transition states.

Intrinsic KIEs, from the chemical step alone, are needed to provide transition state information. Correction of observed to intrinsic KIEs involved measurement of commitment factors associated with the TcdB-GTD GH reaction. Commitment factors measure the probability that molecules in the Michaelis

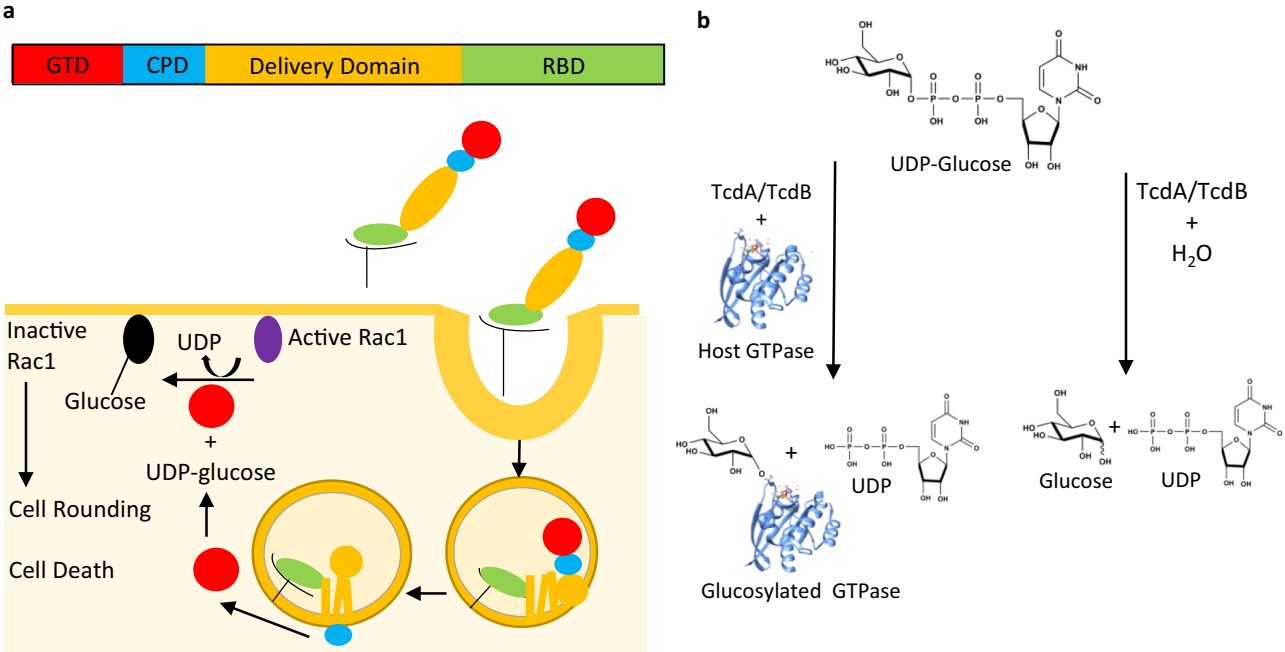

**Fig. 1 Mechanism of action of *C. difficile* TcdA and TcdB. a** (top) Schematic of the Tcd toxin domain structure: N-terminal glucosyltransferase domain GTD (red), cysteine protease domain CPD (blue), delivery domain (orange) and receptor binding domain RBD (green). (Bottom) Schematic of mechanism of TcdA and TcdB-induced toxicity of mammalian cells (see "Introduction" for details), adapted from reference 25. **b** Schematic of TcdA-GTD and TcdB-GTD catalyzed reactions. Glucosyltransferase reaction catalyzed by TcdB-GTD and TcdA-GTD (left). Glucohydrolase reaction catalyzed by TcdB-GTD and TcdA-GTD with water acting as the nucleophile (right).

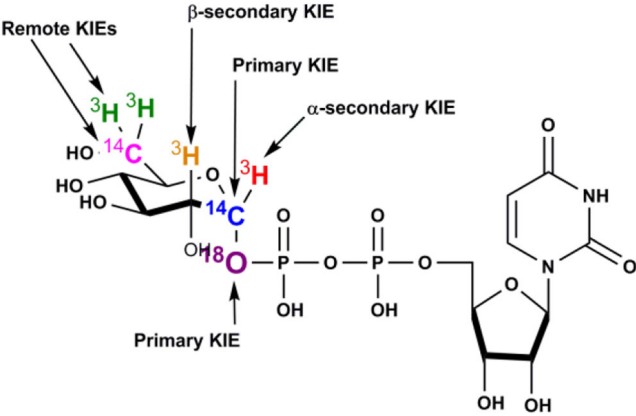

**Fig. 2 KIEs measured for TcdA-GTD and TcdB-GTD glucohydrolase reaction.** Chemical structure of UDP-glucose. Atoms that were labeled for KIE measurements are colored and the type of KIE is indicated.

complex will partition to product or be released to the unreacted substrate pool.

The forward commitment factor for the TcdB-GTD GH reaction was measured using the substrate trapping method developed by Rose[34]. The forward commitment factor $C_f$ was measured to be $0.064 \pm 0.010$ and was used to correct the measured KIEs for TcdB-GTD (Supplementary Fig. 1). The TcdA GH reaction is slower than the TcdB GH reaction[27], and therefore the measured KIEs for TcdA-GTD were assumed to be intrinsic.

UDP-glucose was synthesized enzymatically or chemoenzy-matically from appropriately labeled glucose as starting material to yield $[1''\text{-}^3\text{H}]$, $[2''\text{-}^3\text{H}]$, $[1''\text{-}^{14}\text{C}]$, $[1''\text{-}^{18}\text{O}, 6''\text{-}^{14}\text{C}]$, $[6''\text{-}^3\text{H}]$ and $[6''\text{-}^{14}\text{C}]$UDP-glucose substrates (Fig. 2). As $[6''\text{-}^{14}\text{C}]$ is a heavy atom 3 bonds away from the anomeric carbon, the KIE was

assumed to be negligible and served as the reference. Therefore, $1''\text{-}^3\text{H}$, $2''\text{-}^3\text{H}$ and $6''\text{-}^3\text{H}$ KIEs were measured in pairs with $[6''\text{-}^{14}\text{C}]$UDP-glucose serving as the control (Table 1). KIEs for $1''\text{-}^{14}\text{C}$, $1''\text{-}^{18}\text{O}$, $6''\text{-}^{14}\text{C}$ and $6''\text{-}^{14}\text{C}$ were measured in pairs with $[6''\text{-}^3\text{H}]$UDP-glucose serving as the remote label, and corrected for remote effects by the latter (Table 1). The $1''\text{-}^{14}\text{C}$ KIE reports on bonding and reaction coordinate motion of the anomeric carbon at the transition state. $1''\text{-}^{14}\text{C}$ KIEs of magnitude 1.00–1.02 are typical for an $S_N1$ mechanism with a discrete glucocation intermediate, 1.025–1.06 for an $S_N1$-like dissociative concerted mechanism and 1.06–1.16 for an $S_N2$ associative concerted mechanism[35]. The $1''\text{-}^{14}\text{C}$ KIE was $1.045 \pm 0.001$ for the TcdB-GTD and $1.029 \pm 0.003$ for TcdA-GTD. The KIE results for both enzymes support dissociative $S_N1$-like transition states with considerable glucocation character. TcdA has the more developed glucocation at the transition state.

The α-secondary $1''\text{-}^3\text{H}$ KIE reports on changes in hybridization at the anomeric carbon (sp$^3$ vs sp$^2$) reflecting the degree of glucocation character at the transition state[31,35]. The $1''\text{-}^3\text{H}$ KIE was measured to be $1.216 \pm 0.001$ for TcdB-GTD and $1.261 \pm 0.002$ for TcdA-GTD. These large KIEs are also indicative of a glucocation transition state due to change in hybridization of the anomeric carbon from sp$^3$ to sp$^2$. The sp$^2$ hybridization at the transition state creates enhanced out-of-plane bending freedom of the hydrogen atom, resulting in a large KIE[31,35]. In concert with the $1''\text{-}^{14}\text{C}$ KIEs, TcdA has a more developed glucocation than TcdB.

The β-secondary $2''\text{-}^3\text{H}$ KIE reports on the degree of hyperconjugation that occurs from the σ(C–H) orbital at C2'' to the σ*(C–O) orbital from the anomeric carbon to the UDP leaving group[35,36]. The $2''\text{-}^3\text{H}$ KIE for TcdB-GTD was measured to be $1.014 \pm 0.001$ and $1.052 \pm 0.005$ for TcdA-GTD. The magnitude of both $2''\text{-}^3\text{H}$ KIEs suggests that the C2''–H2'' bond is near-perpendicular to the C1''-UDP bond at the transition state[36]. Other N-glycohydrolases and glycosyltransferases with

glucocation-like transition states express β-secondary 2″-³H KIEs >1.07[35,37]. As the TcdB-GTD and TcdA-GTD 1″-¹⁴C and 1″-³H KIEs support a glucocationic transition state, the 2″-³H GH KIEs indicate an unusual transition state geometry with a lack of hyperconjugation from the σ(C-H) orbital at C2″. Glucocationic transition states are expected to generate C5″–O5″–C1″–C2″ atoms of the glucose ring in a near-co-planar geometry. Four possible conformations of the hexopyranose ring can accommodate the co-planar arrangement, $^{2,5}B$, $B_{2,5}$, $^{3}H_{4}$ and $^{4}H_{3}$[38,39]. Of the four possible structures, the $B_{2,5}$ hexopyranose structure supports small β-secondary KIEs, where the C2″-H2″ bond is nearly perpendicular to the breaking C1″-UDP bond with little hyperconjugation.

The 1″-¹⁸O KIE also reports on the extent of C1″-UDP bond loss at the transition state and was 1.027 ± 0.002 for TcdB-GTD and 1.049 ± 0.005 for TcdA-GTD[35]. A fully cleaved C-O bond to the UDP leaving group is expected to generate a maximum anionic leaving group KIE of 1.047[40]. Therefore, the maximal 1″-¹⁸O KIE for TcdA-GTD indicates that the C1′-UDP bond is broken at the transition state. For TcdB-GTD, the 1″-¹⁸O KIE measured in this study could reflect partial cleavage of the glycosidic bond. Alternatively, leaving group ¹⁸O KIEs for acid-catalyzed C-O bond cleavage of sugar glucosides are thought to proceed through fully protonated transition states and are reported to give KIEs of 1.023–1.026[41,42]. The 1″-¹⁸O KIE for TcdB-GTD could be interpreted as either partial bond cleavage with minimal protonation of the leaving group or extensive bond cleavage at the transition state with protonation of the leaving group oxygen.

A remote 6″-³H KIE was measured to be 1.056 ± 0.001 for TcdB-GTD and 1.065 ± 0.003 for TcdA-GTD. Remote tritium binding isotope effects contribute to the V/K isotope effects, and commonly arise from enzyme-induced distortion of the sp³ geometry at C6″, which is common among other GH enzymes[33,35]. Together these KIEs support the formation of a glucocation-like transition state for both TcdB-GTD and TcdA-GTD GH reactions. The slightly smaller 1″-¹⁴C and slightly larger 1″-³H KIEs measured for the TcdA-GTD reaction suggests that the TcdA-GTD GH transition state exhibits higher glucocation character compared to TcdB-GTD.

**Glucocation transition state analogue inhibitors of TcdA and TcdB.** Features of the transition states for the TcdB-GTD and TcdA-GTD GH reactions include the development of positive charge centered around the anomeric carbon, extending toward the endocyclic oxygen, and a flattened glucose ring through atoms C5″-O5″-C1″-C2″. We tested small molecules that mimic the

GH transition state features for inhibition of TcdB-GTD (Fig. 3 and Table 2). Potential transition state mimics were tested for inhibition of both GH and GT activity, a demonstration for inhibition at the UDP-glucose site rather than at the Rho GTPase binding site (Rac1). Gluconolactone, a well described GH inhibitor contains an sp² hybridized carbon at the anomeric position, mimicking GH transition state geometry, but not charge[43]. Gluconolactone is a weak inhibitor of TcdB-GTD GH activity with an inhibition constant $K_i = 414 \pm 45\,\mu M$ and GT activity with a $K_i = 440 \pm 34\,\mu M$.

Iminosugars contain an endocyclic nitrogen atom that is protonated at physiological pH, mimicking glucocationic transition states[44,45]. Natural product analogues, deoxynojirimycin, isofagomine and noeuromycin are known inhibitors of α and β-glucosidases[45]. The endocyclic nitrogen atom is located at the position of the endocyclic oxygen atom for deoxynojirimycin and at the anomeric carbon for isofagomine and noeuromycin[45]. Deoxynojirimycin did not inhibit TcdB-GTD GH or GT activity at the highest concentration tested (10 mM). Isofagomine and noeuromycin inhibited TcdB-GTD GH activity with $K_i = 4.8 \pm 0.2\,\mu M$ and $K_i = 12.0 \pm 1.6\,\mu M$, respectively. Inhibition of TcdB-GTD GT activity gave $K_i = 1.4 \pm 0.1\,\mu M$ and $K_i = 10.6 \pm 1.0\,\mu M$, respectively. The protonated nitrogen atom at the anomeric position is preferred for TcdB-GTD as deoxynojirimycin is inactive. Although isofagomine lacks the C2″ hydroxyl group of glucose, present in noeuromycin, the C2″ hydroxyl group does

**Table 1 Experimental and intrinsic KIEs of TcdA-GTD and TcdB-GTD glucohydrolase reaction.**

| Label | Type of KIE | Experimental KIE TcdB-GTD | Intrinsic KIE TcdB-GTD | Intrinsic KIE TcdA-GTD |
|---|---|---|---|---|
| 1″-³H | α-secondary | 1.203 ± 0.001 | 1.216 ± 0.001 | 1.261 ± 0.002 |
| 1″-¹⁴C | Primary label | 1.042 ± 0.001 | 1.045 ± 0.001 | 1.029 ± 0.001 |
| 1″-¹⁸O | Primary leaving group | 1.026 ± 0.002 | 1.027 ± 0.002 | 1.049 ± 0.002 |
| 2″-³H | β-secondary | 1.012 ± 0.001 | 1.014 ± 0.001 | 1.052 ± 0.005 |
| 6″-³H | Remote label | 1.053 ± 0.001 | 1.056 ± 0.001 | 1.065 ± 0.003 |
| 6″-¹⁴C | Remote label | 1.000 | 1.000 | 1.000 |

Positions of isotopic labels in the substrate UDP-glucose designed for multiple KIE measurement and the resulting experimental KIEs.
Six independent measurements were performed for each KIE and the data represent the mean values ± SEM.
Experimental KIEs were corrected for forward commitment using Eq. 5 to give intrinsic KIE values.
The 6″-¹⁴C value of unity provides the standard unity reference for all other isotope effects.
Source Data are provided as a Source Data file.

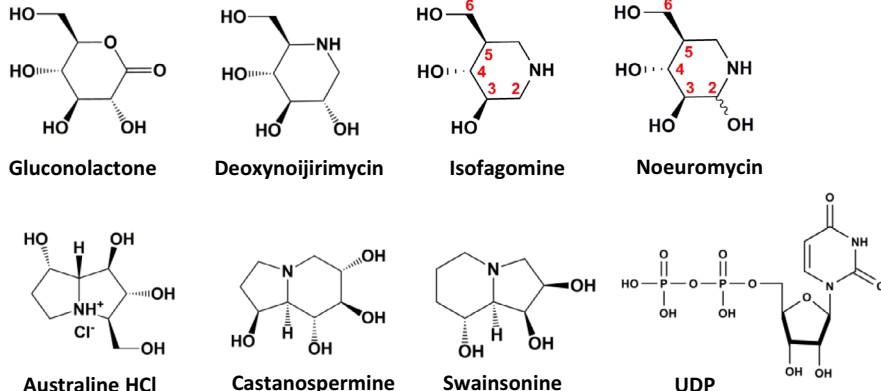

**Fig. 3 Inhibition candidates for TcdA-GTD and TcdB-GTD catalytic activity.** Chemical structures of transition state analogues that were tested for inhibition of TcdA-GTD and TcdB-GTD in this study. Positions of relevant carbon atoms in isofagomine and noeuromycin are indicated.

**Table 2 Inhibition data for TcdB-GTD and TcdA-GTD glucohydrolase and glucosyltransferase activity.**

| Inhibitor | $K_i$ TcdB-GTD glucohydrolase activity (µM) | $K_i$ TcdB-GTD glucosyltransferase activity (µM) | $K_i$ TcdA-GTD glucosyltransferase activity (µM) |
|---|---|---|---|
| Gluconolactone | 413.6 ± 45.5 | 439.7 ± 34.4 | – |
| Deoxynojirimycin | NI | NI | – |
| Isofagomine | 4.8 ± 0.2 | 1.4 ± 0.1 | 0.24 ± 0.01 |
| Noeuromycin | 12.0 ± 1.6 | 10.6 ± 1.0 | 4.7 ± 0.2 |
| Australine HCl | NI | 1570 ± 430 | – |
| Castanospermine | 3400 ± 200 | 680 ± 20 | – |
| Swainsonine | NI | NI | – |
| UDP | – | 11.5 ± 2.9 | 81.1 ± 3.1 |
| Isofagomine (2 x $K_i$ UDP) | – | 0.29 ± 0.06 | 0.016 ± 0.002 |
| Noeuromycin (2 x $K_i$ UDP) | – | 4.4 ± 0.1 | 0.214 ± 0.055 |

Inhibition constants ($K_i$) for transition state analogues tested for inhibition of GH and GT activity of TcdB-GTD and TcdA-GTD.
NI: No inhibition ($K_i > 10$ mM)
The data represents the mean value of 3 independent assays ± SEM
Source data are provided as a Source Data file.

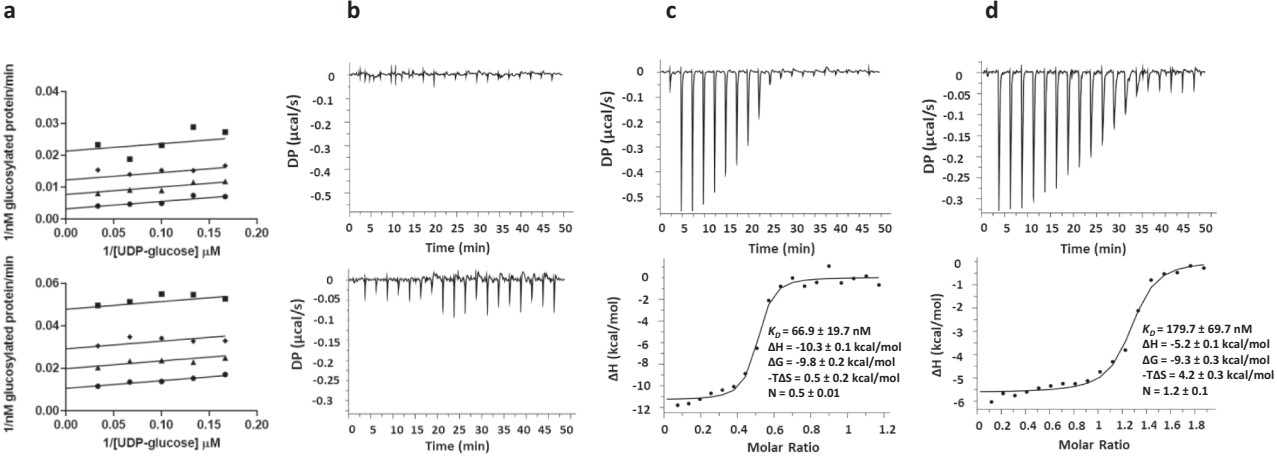

**Fig. 4 Isofagomine and noeuromycin binding to TcdB-GTD. a** Double reciprocal plots are shown for isofagomine (top) and noeuromycin (bottom). TcdB-GTD activity was measured in the presence of inhibitor and varying concentrations of UDP-glucose. Isofagomine or noeuromycin; 0 µM, circles; 2 µM isofagomine or 16 µM noeuromycin, triangles; 4 µM isofagomine or 32 µM noeuromycin, diamonds; 8 µM isofagomine or 64 µM noeuromycin Squares. Data points are the mean of experimental data from 3 independent experiments and lines represent the global fit to Eq. 12 for uncompetitive inhibition as described in Methods. **b** Isothermal titration calorimetry (ITC) analysis of isofagomine (top) or noeuromycin (bottom) binding to TcdB-GTD. **c** ITC analysis of isofagomine binding to TcdB-GTD in the presence of UDP. **d** ITC analysis of noeuromycin binding to TcdB-GTD in the presence of UDP. Conditions are provided in Methods. Top panels indicate raw heat data, and bottom panels show integrated heat injections which have been normalized per mole of injectant as a function of molar ratio. Results were best fit to a one-site binding model. Binding parameters represent mean ± SEM for three replicates. Source data are provided as a Source Data file.

not improve inhibition. In other glucosidases, the C2″ hydroxyl group of glucose contributes considerably to transition state stabilization[44]. Here, noeuromycin was 7.6-fold less potent than isofagomine, thus the C2″ hydroxyl group does not improve inhibitory activity.

Natural product australine did not inhibit TcdB-GTD GH ($K_i > 10$ mM), and showed weak inhibition of TcdB-GTD GT activity with $K_i = 1570 \pm 430$ µM. Castanospermine showed weak inhibition of TcdB-GTD GH activity with $K_i = 3400 \pm 200$ µM and was a weak inhibitor of TcdB-GTD GT activity with $K_i = 680 \pm 20$ µM. Castanospermine has been reported to inhibit both *C. sordelli* lethal toxin and TcdB-GTD with IC$_{50}$ values of approximately 100 µM and 400 µM, respectively[46]. Swainsonine showed no inhibition of TcdB-GTD GH or GT activity ($K_i > 10$ mM). Isofagomine and noeuromycin were the most potent TcdB and TcdA inhibitors. Isofagomine and noeuromycin inhibited TcdA-GTD GT activity with $K_i = 0.24 \pm 0.01$ µM and $K_i = 4.7 \pm 0.2$ µM respectively, confirming inhibition of both Tcd toxins.

The inhibition constants for isofagomine and noeuromycin were 5.8-fold and 2.3-fold improved for TcdA-GTD respectively relative to TcdB-GTD. The more powerful inhibition of TcdA-GTD GH is consistent with its more developed glucocation character at the transition state.

**Isofagomine and noeuromycin form ternary complexes with UDP on TcdA and TcdB.** The GT activity of TcdB-GTD was measured in the presence of varying concentrations of isofagomine or noeuromycin and UDP-glucose. Double reciprocal inhibition plots versus 1/UDP-glucose concentration gave uncompetitive patterns for both inhibitors, indicative of inhibitor binding to form a ternary complex (Fig. 4a and Supplementary Fig. 2). The logical complex for iminosugar binding is to the enzyme–UDP complex. To support this, isofagomine and noeuromycin showed no significant binding to TcdB-GTD alone as indicated by isothermal titration calorimetry (ITC) (Fig. 4b). ITC analysis of UDP binding to TcdB-GTD gave a dissociation

constant $K_D = 49.4 \pm 4.2\,\mu M$ with favorable enthalpic ($\Delta H = -14.3 \pm 3.9$ kcal/mol) and unfavorable entropic ($-T\Delta S = 8.2 \pm 3.8$ kcal/mol) contributions (Supplementary Fig. 3). Isofagomine and noeuromycin titration with TcdB-GTD in the presence of near-saturating concentrations of UDP gave readily measurable binding constants (Figs. 4c, d). With UDP, isofagomine and noeuromycin gave $K_D = 0.067 \pm 0.019\,\mu M$ and a $K_D = 0.180 \pm 0.070\,\mu M$ respectively. Isofagomine and noeuromycin binding is enthalpically driven with $\Delta H = -10.3 \pm 0.1$ kcal/mol and $\Delta H = -5.2 \pm 0.1$ kcal/mol, respectively. The entropic contribution for isofagomine and noeuromycin binding was $-T\Delta S = 0.5 \pm 0.2$ kcal/mol and $-T\Delta S = 4.2 \pm 0.3$ kcal/mol, respectively. Larger enthalpic contributions of isofagomine binding to TcdB-GTD in the presence of UDP is consistent with the $K_i$ values presented in Table 2, making isofagomine the more potent inhibitor. Isofagomine and noeuromycin binding to TcdB-GTD relies on the presence of UDP, suggesting UDP organization of the catalytic site and/or isofagomine and noeuromycin interaction with UDP in the active site. Furthermore, the results from the ITC binding studies reveal ordered product release, with glucose released before UDP. This is consistent with reports of ordered product release for TcdA-GTD[47].

Kinetic assays for TcdB-GTD and TcdA-GTD GT activity with a fixed concentration of UDP at 2 x $K_i$ concentration gave improved inhibition constants. For TcdB-GTD, $K_i = 0.29 \pm 0.06\,\mu M$ and $4.4 \pm 0.1\,\mu M$ for isofagomine and noeuromycin, respectively (Table 2). Inhibitory affinity was improved by 2.4-fold and 4.8-fold respectively in the presence of UDP. Inhibition constants for isofagomine and noeuromycin against TcdA-GTD GT activity were $K_i = 16 \pm 2$ nM and $K_i = 210 \pm 50$ nM, respectively; an improvement of 15- and 23-fold respectively in the presence of UDP. Improved inhibitory activity against TcdA-GTD supports the KIE measurements. Isofagomine and noeuromycin inhibit both toxins through binding to form a ternary complex in the presence of UDP.

**Structures of TcdB-GTD in complexes with UDP, isofagomine or noeuromycin**. The geometry of UDP, isofagomine and noeuromycin bound to TcdB-GTD was established by solving the crystal structures. The structural fold of the TcdB-GTD-inhibitor complexes is the same as the TcdB-GTD structures as reported previously (Supplementary Fig. 4)[48,49]. Electron densities were clearly observed for the entire polypeptide, except for a few surface side chain residues. Electron densities for UDP, isofagomine and noeuromycin were also well resolved in the crystal structures (Supplementary Fig. 5). The structure of TcdB-GTD in complex with UDP and isofagomine was solved at 1.82 Å in the space group $P2_12_12_1$ with two monomers in the asymmetric unit (PDB ID: 7LOU, Supplementary Fig. 4, Supplementary Table 1). The binding of UDP and isofagomine is well resolved in the binding pocket (Fig. 5a). The uracil ring is π-π stacked with Trp102 and sandwiched between the side chains of Trp102 and Leu265. O2 and O4 of the uracil ring are hydrogen bonded with the backbone amine of Ile103 and the amide of Asn139, respectively, while N3 interacts with the backbone carbonyl of Ile103. The 2′-hydroxyl of the UDP ribose is hydrogen bonded with the Val101 carbonyl and Ser269 hydroxyl, whereas the 3′-hydroxyl hydrogen bonds with the Asp286 carboxyl and a water molecule. The α-phosphate of UDP forms hydrogen bonds to the backbone amine of Leu519 and the carboxyl of Asp288. The α and β phosphates of UDP also interact with structural waters and coordinate to the $Mn^{2+}$ metal cation, which is in turn coordinated to the carboxyl groups of Asp288 and Glu515 (Fig. 5a). The β-phosphate of UDP is positioned by hydrogen bonds with the Ser518 hydroxyl and the aromatic N of Trp520. The cationic nitrogen of isofagomine is

2.6 Å from O of the β-phosphate of UDP in an ion-pair interaction. A structural water molecule is 2.7 Å on the opposite face of the isofagomine cation in appropriate position to act as the attacking nucleophile for the hydrolytic reaction of TcdB with UDP-glucose. O3 of isofagomine is hydrogen bonded with the carboxylate of Asp286 and the Arg273 guanidinium. The O4 and O6 of the isofagomine share H-bond interactions with both carboxylate oxygens of Asp270 (Fig. 5a).

The TcdB-GTD complex with UDP-noeuromycin was solved at 2.50 Å resolution in the C2 space group with two TcdB-GTD molecules in the asymmetric unit (PDB ID: 7LOV, Supplementary Fig. 4, Supplementary Table 1). The binding of UDP and noeuromycin is similar to UDP and isofagomine interactions. However, the ion-pair interaction distance between the nitrogen cation of noeuromycin and the β-phosphate oxygen is slightly weaker at 2.7 Å. The C2-hydroxyl of the noeuromycin also interacts (3.0 Å) with a β-phosphate of the UDP (Fig. 5b). Despite the extra interaction of the noeuromycin C2-hydroxyl, the combination of this interaction and the weaker nitrogen cation to UDP interaction together caused reduced inhibition potency when compared to isofagomine (Fig. 5b and Supplementary Fig. 6).

The structures of TcdB-GTD with UDP and isofagomine (PDB ID: 7LOU) or noeuromycin (PDB ID: 7LOV) were compared with apo and UDP-2-deoxy-2-fluoroglucose bound TcdB-GTD and TcdA-GTD structures[49,50]. The protein structures are nearly identical to the UDP-2-deoxy-2-fluoroglucose TcdA-GTD and TcdB-GTD bound structures (RMSD: 0.289 to 0.792 Å) (Supplementary Fig. 4). Apo TcdB-GTD structures have loops (Glu449 to Asp461 and Gln510 to Asp523) in open conformations (Fig. 5c and Supplementary Fig. 6). These loops are closed in the UDP inhibitor-bound structures which is consistent with previously published structures of the Tcd toxins in complex with UDP-glucose and UDP[47,49–51]. The Trp520 residue from a loop moves 10 Å to form a H-bond with the β-phosphate of UDP (Fig. 5c).

**Isofagomine and noeuromycin prevent TcdA- and TcdB-induced cellular toxicity**. Isofagomine and noeuromycin were tested for the ability to prevent TcdA and TcdB-induced toxicity (Fig. 6). Cell rounding is a hallmark of TcdA and TcdB toxicity in mammalian cells and can be visualized by light microscopy. Vero cells were used for quantitation of cell rounding by Tcd toxins in the presence of either isofagomine or noeuromycin (Fig. 6a and Supplementary Fig. 7). The $IC_{50}$ for cell rounding was calculated for both isofagomine and noeuromycin following treatment with either 1 nM TcdA or 1 pM TcdB (Table 3). The $IC_{50}$ values for cell rounding ranged from 5.5 μM to 13.6 μM (Table 3), indicating that both isofagomine and noeuromycin prevent TcdA and TcdB cell rounding in a dose-dependent manner. Isofagomine and noeuromycin also prevented Tcd toxin-induced cell rounding of IMR90 cells (Supplementary Fig. 8). The mechanism of action for isofagomine and noeuromycin protection against Tcd toxins was determined by Western blot analysis of Rac1 glucosylation as described in Tam et al[52]. Rac1 glucosylation used anti-Rac1 antibody (Mab102) which distinguishes between unglucosylated and glucosylated Rac1. IMR90 cells were treated with varying concentrations of isofagomine or noeuromycin (12.5–100 μM) and TcdA (1 nM) or TcdB (0.1 nM). Treatment of IMR90 cells with increasing concentrations of either isofagomine or noeuromycin caused a dose-dependent reappearance of unglucosylated Rac1 (Mab102) as compared to total Rac1 levels (anti-Rac1 23A8) (Fig. 6b and Supplementary Figs. 9, 10). Therefore, both iminosugars prevent toxin-induced cell rounding through inhibition of TcdA and TcdB GT activity. Finally, we measured induction of

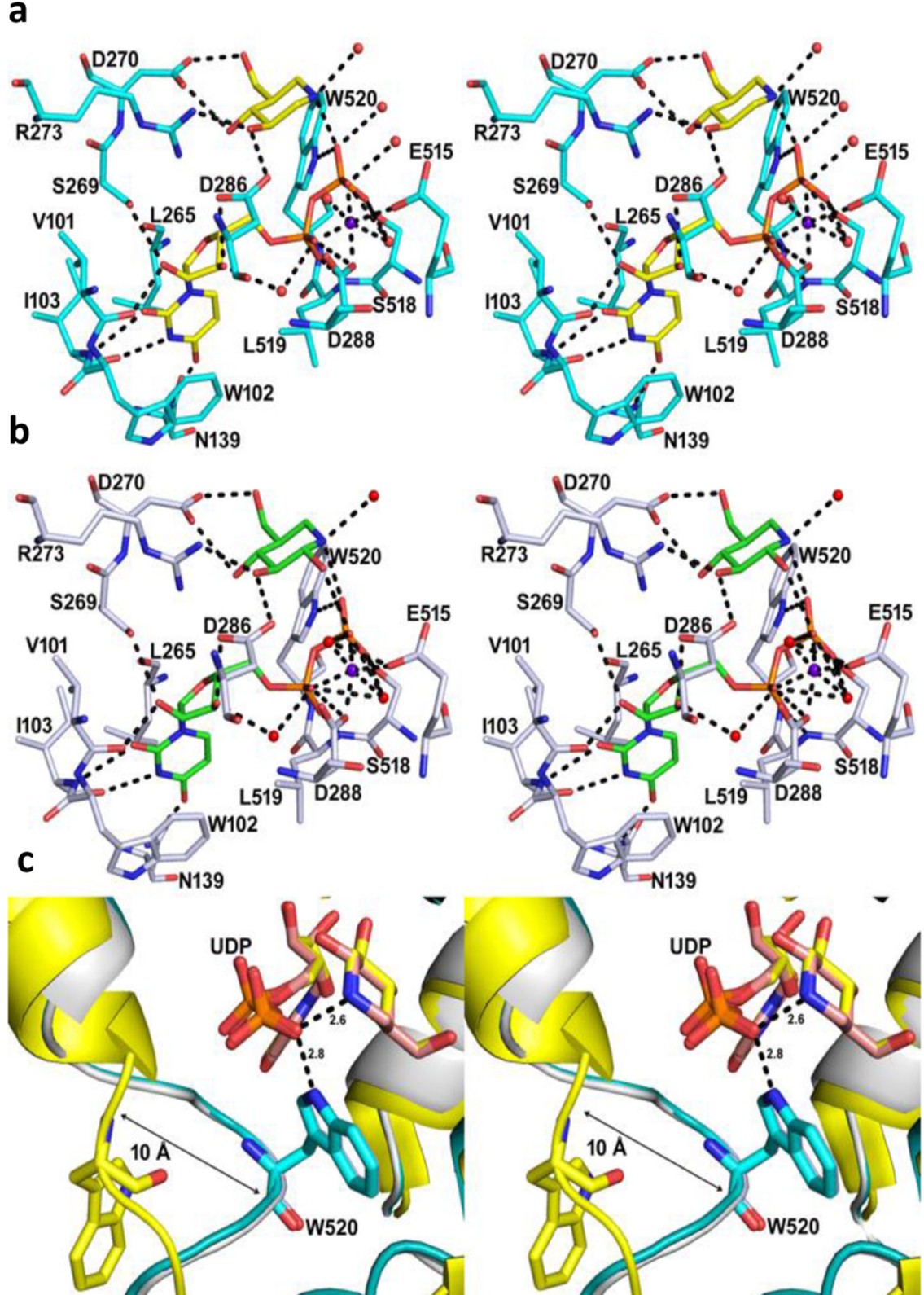

**Fig. 5 Binding interactions of TcdB-GTD in complex with inhibitors (stereo-views).** Inhibitor complexes of isofagomine (PDB ID: 7LOU, yellow) and noeuromycin (PDB ID: 7LOV, green) are shown in panels (**a**) and (**b**), respectively. Amino acid residues interacting with inhibitors and UDP are indicated. Selected hydrogen bond interactions are shown in black dotted lines. The π-π stacking interaction of uridine ring with Trp102 is also highlighted.
**c** Superposition of the binding pocket of apo TcdB-GTD (PDB ID: 5UQT, yellow), isofagomine (PDB ID: 7LOU, cyan), and noeuromycin (PDB ID: 7LOV, gray) bound structures. Movement of the loop from residues Gln510 to Asp523 (10 Å) towards the binding pocket occurs on inhibitor binding and is highlighted with the black arrow.

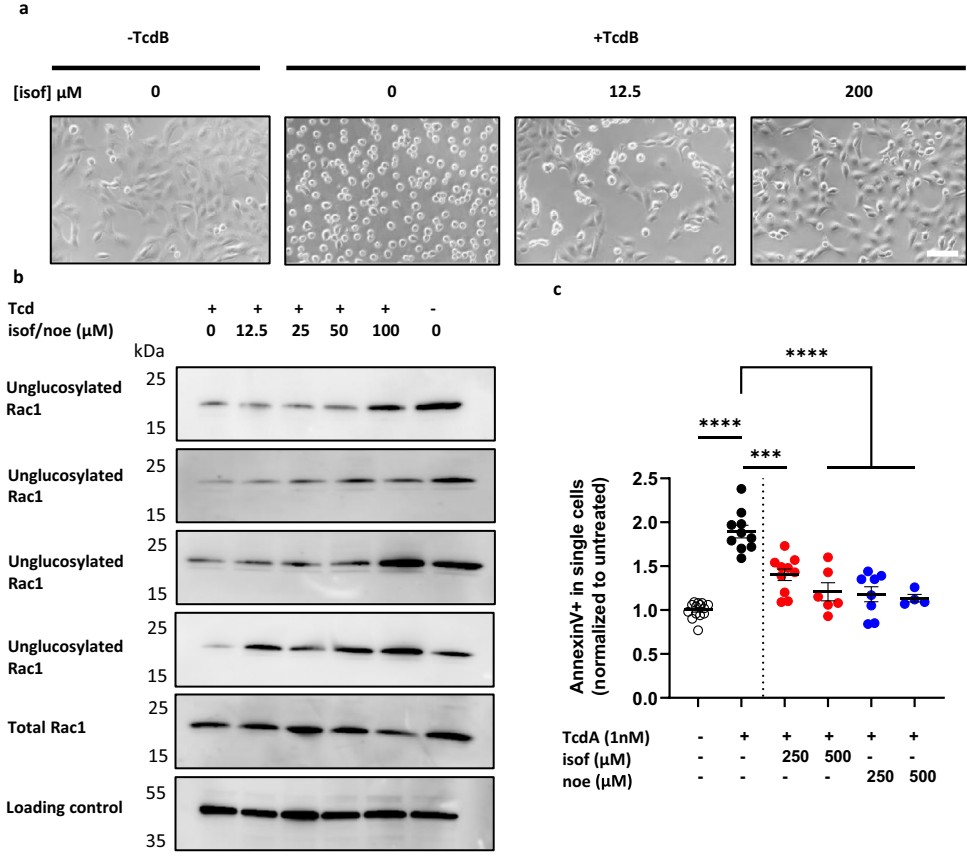

**Fig. 6 Isofagomine and noeuromycin prevent TcdA and TcdB-induced cytotoxicity in mammalian cells. a** Representative image from at least 3 experiments of Vero cells 2 h after treatment with TcdB and isofagomine (isof). Scale bar represents 10 μM. **b** Representative Western blot images for intracellular Rac1 glucosylation (n=3). Human IMR90 cells were treated with isofagomine (isof) or noeuromycin (noe) (doses indicated) for 30 min, followed by treatment with buffer control, 1 nM TcdA or 0.1 nM TcdB for 6 h. Cell lysates were prepared as described in Methods. Mab102 was used to detect unglucosylated Rac1 in cell lysates. Anti-Rac1 (23A8) was used to detect total Rac1 levels and anti-GAPDH served as the loading control. Uncropped Western blots are shown in Supplementary Figs. 9, 10. **c** Flow cytometry analysis of AnnexinV in HT-29 cells from 5 independent experiments (n=5). HT-29 cells were pre-treated with isofagomine (isof) or noeuromycin (noe) (250 μM or 500 μM) for 30 min before treatment with buffer control or 1 nM TcdA for 24 h. Cells were harvested and stained with AnnexinV as described in Methods. AnnexinV positivity (%) was normalized to untreated HT-29 cells. Untreated cells are shown as white circles, TcdA treated cells are shown in black circles, isofagomine and TcdA treated cells are shown as red circles and noeuromycin and TcdA treated cells are shown as blue circles. Ordinary one-way ANOVA with Tukey's multiple comparison test, significance ****$p < 0.0001$, *** $p < 0.001$. Error bars show mean ± SEM. Source data are provided as a Source Data file.

apoptosis in the presence of TcdA and either isofagomine or noeuromycin, based on the protocol described in Lica et al[53]. Annexin V binds to phosphatidylserine which is exposed on the outer side of the cell membrane during early apoptosis. HT-29 cells were incubated with isofagomine or noeuromycin (250 μM or 500 μM) and analyzed for Annexin V signal by flow cytometry at 24 h post-treatment. TcdA induced apoptosis, while addition of either iminosugar prevented apoptosis (Fig. 6c and Supplementary Fig. 11). Thus, isofagomine and noeuromycin prevent Tcd-induced cell rounding of cultured mammalian cells via inhibition of Tcd GT activity and prevent apoptotic cell death in HT-29 cells. Isofagomine and noeuromycin are therefore attractive candidates for protection against TcdA and TcdB cellular toxicity in CDI.

## Discussion

*Clostridium difficile* opportunistic infections usually occur following antibiotic treatment. The normal gut microbiome is disturbed and *C. difficile* repopulates the gut. This is the leading cause of hospital-acquired bacterial infections and is listed as an urgent threat for antibiotic resistance by the CDC[1]. Ideally, blocking the *C. difficile* virulence factors, TcdA and TcdB without

disruption of the gut microbiome is a preferred prevention and treatment strategy. Once TcdA and TcdB enter mammalian cells they are processed in endosomes, releasing the N-terminal glucosyltransferase domain which glucosylates RhoGTPases and disrupts the actin cytoskeleton network. The approval of Bezlotoxumab and related monoclonal antibodies (mAbs) targeting TcdA and TcdB are a step in this direction. However, mAbs access the wrong cellular compartment to be effective against the toxins and there remains an unmet need for orally available small molecule therapeutics. Previous studies have identified small molecule inhibitors that target the Tcd toxins[23–25,52]. However, none to date have focused on the development of transition state analogues targeting the GT domain. Here we used KIEs to probe transition state features of the TcdA-GTD and TcdB-GTD catalyzed hydrolysis of UDP-glucose. KIE analysis of both reactions resulted in an intrinsic KIE pattern consistent with glucocation-like transition states where considerable positive charge develops on the sp2-like anomeric carbon and the sugar ring is flattened through C5″–O5″–C1″–C2″. The KIE analysis of both reactions revealed that the TcdA-GTD transition state has slightly more glucocation character. The TcdA and TcdB glucosyltransferase reactions proceed through retention of stereochemistry at the

**Table 3 Efficacy of isofagomine and noeuromycin against TcdA and TcdB-induced cell rounding of Vero cells.**

| C. difficile toxin and iminosugar | Calculated IC$_{50}$ (μM) |
|---|---|
| TcdA + Isofagomine | 5.5 ± 1.5 |
| TcdA + Noeuromycin | 9.6 ± 3.3 |
| TcdB + Isofagomine | 8.3 ± 3.6 |
| TcdB + Noeuromycin | 13.6 ± 5.6 |

The data represents the mean of three biological replicates ± SEM.
Source data are provided as a Source Data file.

anomeric carbon and are expected to form an internal return, SNi-like transition state[29,37].

Iminosugars isofagomine and noeuromycin were the most potent inhibitors of the toxins and exhibited uncompetitive mechanisms of inhibition. ITC analysis and x-ray crystal structures supported a mechanism where the inhibitors occupy the glucose binding pocket in the presence of UDP, similar to the position of UDP-glucose bound to the toxins. Isofagomine and noeuromycin also prevented cell toxicity induced by full-length TcdA and TcdB. Full structural analogues of the transition state would contain both a UDP and glucocation mimic, however, the presence of a diphosphate moiety precludes cell permeability[54]. Glucosyltransferase bisubstrate analogue inhibitors that contain UDP-glucose donor and/or acceptor components have been synthesized and reviewed in Izumi et al[55]. The linker structure and the length linking the components influence the inhibitory activity. The relatively tight binding of both isofagomine and noeuromycin make them both potential candidates to protect the gut from TcdA and TcdB.

As potential candidates for treatment of CDI, the pharmacokinetic properties of isofagomine and noeuromycin are significant. Isofagomine has been investigated as a potential therapy for Gaucher disease. Isofagomine stabilizes mutant glucocerebrosidase (a β-glucosidase) in an active conformation, acting as a chaperone to maintain an active state in patients with Gaucher disease[56–58]. Oral isofagomine doses were well tolerated at 225 mg/day, however the majority of patients did not show a significant reduction in Gaucher disease symptoms and therefore isofagomine was withdrawn after phase II clinical trials for this disease (NCTs: 00433147; 00446550; 00813865; 00875160). Isofagomine and noeuromycin have not been previously investigated to treat CDI. Oral doses similar to those in the Gaucher trials are expected to cause gut concentrations well in excess of the cellular IC$_{50}$ values for cellular protection reported here. This study demonstrates significant anti-toxin potential for isofagomine and noeuromycin as small molecule therapeutics to prevent and treat CDI.

## Methods

Isotopically labeled glucoses (1-$^3$H, 2-$^3$H and 6-$^3$H) were purchased from Perkin Elmer and 1-$^{14}$C and 6-$^{14}$C labeled glucose were purchased from American Radiolabeled Chemicals. H$_2$$^{18}$O isotopically labeled water was purchased from Cambridge Isotope Laboratories. Hexokinase, pyruvate kinase, phosphoglucomutase, UDP-glucose pyrophosphorylase and inorganic pyrophosphatase were purchased from Millipore-Sigma. Isofagomine D-tartrate was purchased from Carbosynth. Deoxynojirimycin, castanopermine, swainsonine and australine hydrochloride were purchased from GlycoFine chemicals. D-gluconon-1,5-lactone was purchased from Frontier Scientific. Uridine disphosphoglucose disodium salt (UDP-glucose) was purchased from Abcam. Uridine 5'diphosphate disodium salt hydrate (UDP) was purchased from Millipore-Sigma. Vero, IMR90, CHO-K1 and HT-29 cells were purchased from ATCC (CCL-81, CCL-186, CCL61 and HTB-38). Full-length TcdA and TcdB toxins were purchased from List Biological Laboratories Inc. FITC-Annexin V staining kit was purchased from Biolegend. All other reagents were of analytical grade and purchased from commercial sources.

**Protein expression and purification.** Expression plasmid pET28a containing C-terminal His-6 tagged TcdB-GTD (amino acids 1-546) was kindly provided by Tam et al (ref. [25] in main text). Conditions for the heterologous expression of TcdB-GTD in *Escherichia coli* (One Shot BL21 Star$^{TM}$(DE3) cell line) were adapted from Tam et al. Protein expression was induced by the addition of 0.5 mM IPTG for 3 h at 30 °C. The cells were harvested by centrifugation and resuspended in 50 mL of lysis buffer (20 mM Tris pH 7.5, 5 mM Imidazole and 500 mM NaCl) to which 1 protease inhibitor tablet was added (Roche). Bacterial cells were lysed by sonication with 10 s pulses for 10 min and cell debris was removed by centrifugation at 25,000 × g for 40 min and cleared cell lysate was filtered through 0.45 μm filters. Filtered cell lysate was applied to 5 mL of Ni-NTA resin (Qiagen) which had been previously equilibrated with 10 column volumes of H$_2$O, followed by 10 column volumes of lysis buffer. The resin was washed with 6 column volumes of lysis buffer and followed by 6 column volumes of wash buffer 2 (20 mM Tris pH 7.5, 10 mM imidazole, 500 mM NaCl). TcdB-GTD was eluted using 250 mM imidazole in 20 mM Tris pH 7.5 and 500 mM NaCl. The desired fraction was pooled and exchanged into 20 mM Tris pH 7.5, 150 mM NaCl and glycerol was added to a final concentration of 15% v/v. The protein was stored at −80 °C.

The gene encoding TcdA-GTD (amino acids 1–542) was synthesized with a C-terminal His-6 tag and cloned into pD451-SR:391299 plasmid (Atum Bio). The plasmid was transformed into *E. coli* (One shot BL21 Star$^{™}$(DE3) cell line). Recombinant protein expression and purification was achieved as described for TcdB-GTD.

The gene encoding Rac1 GTPase (amino acids 1-192) was synthesized with a N-terminal His6 tag and cloned into pD454-SR:375415 plasmid (Atum Bio). Rac1 GTPase was expressed in *E. coli* (One shot BL21 Star$^{™}$(DE3) cell line) and recombinant protein expression was induced with 1 mM IPTG for 4 h at 30 °C. Purification of Rac1 was performed as described above with the exception that the lysis buffer, wash buffers and elution buffers contained 300 mM NaCl.

**Synthesis of isotopically labeled substrates.** Isotopically labeled UDP-glucose was synthesized by coupled enzymatic reactions. 1-$^3$H, 2-$^3$H, 6-$^3$H, 1-$^{14}$C and 6-$^{14}$C labeled glucoses were used as starting materials to prepare 1-$^3$H, 2-$^3$H, 6-$^3$H, 1-$^{14}$C and 6-$^{14}$C UDP-glucose respectively. Each synthesis reaction contained 1 mM D-glucose (30 μCi of labeled glucose), 1 mM UTP, 40 μM ATP, 10 mM phosphoenolpyruvate, 50 mM Tris-HCl pH 7.4, 2 mM KH$_2$PO4, 10 mM MgCl$_2$ in a volume of 500 μL. Reactions were initiated by the addition of 1 unit hexokinase, 2 units pyruvate kinase, 1 unit phosphoglucomutase, 2 units of UDP-glucose pyrophosphorylase and 2.5 units of inorganic pyrophosphatase. Reactions were incubated at room temperature overnight and the following day the reactions were passed through 10 kDa cutoff Amicon ultra spin columns to remove enzymes. Isotopically labeled UDP-glucose products were purified from reaction mixtures by anion exchange HPLC using a 1 mL column (GE Mono Q). The column was eluted with a gradient of 0 to 0.5 M ammonium formate and elution was monitored by UV detector at 260 nm. The product was pooled and lyophilized. The product purity was checked by comparing the HPLC peak with commercially available UDP-glucose. 1-$^{18}$O UDP-glucose was prepared from oxygen exchange at the anomeric carbon in isotopically labeled water H$_2$$^{18}$O (Cambridge isotope labs) based on the method by Risley et al[59]. Briefly, 1 mM 6-$^{14}$C glucose was dissolved in 250 μL of H$_2$$^{18}$O containing 5 mM inorganic phosphate pH 7.0. The solution was incubated at 61 °C overnight and lyophilized. The 1-$^{18}$O, 6-$^{14}$C double labeled glucose was then used as the starting material for the synthesis of 1-$^{18}$O, 6-$^{14}$C doubly labeled UDP-glucose, following the same method as described above. For the synthesis of 1-$^{18}$O, 6-$^{14}$C double labeled UDP-glucose, the isotope enrichment was assessed via mass spectrometry on a 'cold' reaction that was run in parallel. Isotope enrichment was assessed to be at least 95%. A sample mass spectrum is presented Supplementary Fig. 12. Mass spectra were acquired on a Shimadzu LCMS-2010EV spectrometer.

**Measurement of V/K KIEs for TcdB-GTD and TcdA-GTD glucohydrolase reaction.** Measurement of V/K KIEs for the TcdB-GTD and TcdA-GTD glucohydrolase reactions were performed at room temperature using the competitive radiolabeled approach. Reaction conditions for TcdB-GTD KIE measurements were: 50 mM HEPES pH 7.0, 100 mM KCl, 4 mM MgCl$_2$, 1 mM MnCl$_2$, 100 μM UDP-glucose ([*heavy label*] UDP-glucose + [*remote label*] UDP-glucose + unlabeled carrier) and 200 nM TcdB-GTD. For TcdA-GTD KIE measurements the reaction conditions were the same as above with the exception of 50 mM HEPES pH 7.5 and 2 μM TcdA-GTD. For a typical KIE measurement, a master mix containing all reaction components (except enzyme) was prepared and equal-volume samples were aliquoted into tubes and designated as 'control reactions' which include the no enzyme control and 3 replicates of the starting heavy/remote label ratio. TcdB-GTD or TcdA-GTD was added to the remaining master mix and the reaction was allowed to proceed to 10-20% completion and then equal-volume aliquots of the TcdB-GTD or TcdA-GTD reaction were added to the same volume of 100 mM EDTA pH 8.0 to quench the reaction. Glucose product was purified using 0.5 mL columns containing Dowex® 1×2 anion exchange resin (Millipore-Sigma) pre-equilibrated with 10 column volumes of H$_2$O and 10 column volumes of 50 mM D-glucose. Radiolabeled glucose was eluted with 1 mL of 50 mM D-glucose and collected in 20 mL liquid scintillation vials and 10 mL of Perkin Elmer Ultima Gold liquid scintillation fluid was added to each scintillation vial.

Scintillation counting was performed on each sample over 10 cycles at 10 min/cycle using a Tri-carb 2910TR scintillation counter (Perkin Elmer), which is a dual-channel instrument that registers the signal for $^3$H in Channel A and the signal for $^{14}$C in both Channel A and Channel B. The raw data was analyzed to determine the total counts for $^3$H-labeled glucose and $^{14}$C-labeled glucose. A control sample of $^{14}$C-labeled glucose was used as a standard to establish the signal overlap between Channel A and Channel B for $^{14}$C as defined by Eq. 1:

$$r = \text{Channel A}/\text{Channel B} \tag{1}$$

Spectral deconvolution of the KIE data was achieved for $^3$H and $^{14}$C using Eqs. 2 and 3 respectively:

$$^3\text{H} = \text{Channel A} - (\text{Channel B} \times r) \tag{2}$$

$$^{14}\text{C} = \text{Channel B} + (\text{Channel B} \times r) \tag{3}$$

$V/K$ KIE values were calculated from Eq. 4, where $R_0$ and $R_f$ are the ratios of [heavy label] UDP-glucose to [remote label] UDP-glucose prior to the reaction and at partial conversion respectively and, $f$ is the fraction of substrate conversion:

$$\text{KIE}_{v/K} = \frac{\text{Log}(1-f)}{\text{Log}\left(1 - f\frac{R_f}{R_0}\right)} \tag{4}$$

For KIE measurements involving 1-$^{14}$C, 1-$^{18}$O and 6-$^{14}$C UDP-glucose. The observed KIE values were corrected for the 6-$^3$H remote isotope effect using Eq. 5:

$$\text{KIE} = \text{KIE observed} \times 6 - {}^3\text{H KIE} \tag{5}$$

**Measurement of TcdB-GTD glucohydrolase forward commitment.** Forward commitment for UDP-glucose in the TcdB-GTD glucohydrolase reaction was measured using the isotope trapping method[34]. A 50 µL sample of TcdB-GTD:UDP-glucose equilibrium complex was formed by incubating 10 µM TcdB-GTD with 100 µM 6-$^{14}$C UDP-glucose for 5 s in 50 mM HEPES pH 7.0, 100 mM KCl, 4 mM MgCl$_2$, and 1 mM MnCl$_2$ at room temperature. A chase solution (450 µL 5 mM unlabeled UDP-glucose, 50 mM HEPES pH 7.0, 100 mM KCl, 4 mM MgCl$_2$, and 1 mM MnCl$_2$) was rapidly mixed with the ES complex solution and 4 × 50 µL aliquots were removed from the mixture and quenched with 50 µL of 100 mM EDTA pH 8.0 at 15, 30, 45 and 60 s after addition of chase solution. Control reactions containing no enzyme were processed in parallel to correct for background levels of UDP-glucose breakdown at room temperature. Labeled glucose product was purified as described above for KIE measurements using columns containing 0.5 mL Dowex® 1×2 anion exchange resin. 10 mL of Ulitma Gold™ scintillation fluid (Perkin Elmer) was added to each sample and scintillation counting was performed as described for KIE measurements. The amount of glucose produced after addition of chase solution was plotted as a function of time and extrapolated to time = 0. The concentration of enzyme–substrate (ES) complex was calculated using Eq. 6, where, $E$ represents the enzyme concentration, $S$ represents the concentration of 6-$^{14}$C UDP-glucose and $K_M$ is the Michaelis constant for UDP-glucose, TcdB-GTD glucohydrolase reaction.

$$ES = (ES)/(S + K_M) \tag{6}$$

$C_f$ was calculated using Eq. 7 where $Y$ is the ratio of moles of glucose product $P$ ($y$-intercept at $x$=0) to moles of the TcdB-GTD:UDP-glucose $ES$ complex (Eq. 8).

$$C_f = Y/(1-Y) \tag{7}$$

$$Y = P/ES \tag{8}$$

For isotope $x$, the intrinsic KIE on an enzymatic reaction ($^x k$) can be extracted from the $^x V/K$ KIE using Northrop's equation (Eq. 9)[32], when forward commitment ($C_f$), reverse commitment ($C_r$) and the equilibrium isotope effect ($^x K_{eq}$) are known.

$$^x V/K = \frac{^x k + C_f + C_r\,^x K_{eq}}{1 + C_f} \tag{9}$$

The TcdB-GTD GH reaction is irreversible under the conditions used to measure KIEs and therefore, $C_r$ is zero under these experimental conditions. Therefore, Eq. 9 reduces to Eq. 10 whereby, $^x k$ can be extracted from the $^x V/K$ KIE using $C_f$ alone:

$$^x V/K = \frac{^x k + C_f}{1 + C_f} \tag{10}$$

**TcdB-GTD and TcdA-GTD glucosyltransferase assays.** TcdB-GTD and TcdA-GTD glucosyltransferase activity was measured using 6-$^3$H UDP-glucose as a substrate and capturing radiolabeled glucosylated Rac1 protein by precipitating the protein product based on the method described in Bensadoun and Weinstein[60]. Briefly, a glucosyltransferase reaction was carried out for at least 10 min at room temperature in a 50 µL reaction mix containing 50 mM HEPES pH 7.0, 100 mM KCl, 4 mM MgCl$_2$, 1 mM MnCl$_2$ 10 µM 6-$^3$H UDP-glucose, 90 µM UDP-glucose, 20 µM Rac1 (for TcdB-GTD) or 50 µM Rac1 (for TcdA-GTD) and purified TcdB-

GTD or TcdA-GTD at an appropriate concentration to detect the completed reaction in the linear phase (typically 1 nM). At the completion of the assay reactions were terminated with the addition of 50 µL of 100 mM EDTA pH 8.0. The terminated reaction samples (100 µL) were added to a precipitation mix containing 120 µg/mL sodium deoxycholate, 6% trichloroacetic acid and 10 µg/mL BSA. Samples were incubated at room temperature for 15 min before being centrifuged for 20 min at 15,000 × g to separate the glucosylated protein product. The supernatant was removed and the protein pellet was resuspended in 500 µL of 200 mM Tris-HCl pH 7.5, 5% SDS and 20 mM NaOH. Samples were vortexed briefly and added to 20 mL scintillation vials to which 10 mL of Ultima Gold scintillation fluid (Perkin Elmer) was added. The amount of radioactivity in each sample was measured in a PerkinElmer Tricarb 2910 TR scintillation counter. For inhibition assays quantification of glucosylated Rac1 was performed as described above, with exception of 15 µM total UDP-glucose concentration (7.5 µM 6-$^3$H UDP-glucose) and 20 µM or 50 µM Rac1 for reactions containing TcdB-GTD and TcdA-GTD respectively. The IC$_{50}$ value of each compound was determined from a dose–response curve by varying the concentration of the inhibitor under the same enzyme concentration. The data were analyzed with GraphPad Prism software using a non-linear fit of log$_{10}$ [inhibitor] vs. normalized response. The Inhibition constant $K_i$ for each compound was calculated using Eq. 11 [61] after confirming the mode of inhibition using Lineweaver-Burk Plots as described below. S represents the total substrate concentration (10 µM for assays with TcdB-GTD and 15 µM for assays with TcdA-GTD) and $K_M$ is the Michaelis constant for UDP-glucose (21.3 µM for TcdB-GTD and 13.8 µM for TcdA-GTD).

$$K_i = \frac{\text{IC}_{50}}{1 + \frac{[S]}{[K_M]}} \tag{11}$$

The mode of inhibition was investigated by varying the concentration of inhibitor and varying concentrations of UDP-glucose. The data was plotted as double reciprocal plots and assessed using Lineweaver-Burk analysis. Data were fit to Eq. 12 for uncompetitive inhibition or Eq. 13 for competitive inhibition using SigmaPlot 6.0 where v is the measured reaction velocity, $V$ is the maximal velocity, $A$ is the concentration of substrate (UDP-glucose), $K_a$ is the corresponding Michaelis–Menten constant, $I$ is the concentration of inhibitor and $K_{is}$ and $K_{ii}$ are the slope and intercept inhibition constants for the inhibitor respectively. Data reported here are the mean of three independent assays.

$$v = VA/\left[K_a + A\left(1 + \frac{I}{K_{ii}}\right)\right] \tag{12}$$

$$v = VA/\left[K_a\left(1 + \frac{I}{K_{is}}\right) + A\right] \tag{13}$$

**Synthesis of isofagomine and noeuromycin.** All experimental details for the synthesis and characterization of isofagomine and noeuromycin are provided in the supplementary material.

**Isothermal titration calorimetry experiments.** All ITC experiments were conducted on a Microcal PEAQ-ITC (Malvern Instruments). The experiments were performed at 25 °C in a cell containing 280 µL of reaction mixture (50 mM HEPES pH 7.5, 100 mM KCl, 4 mM MgCl$_2$, 1 mM MnCl$_2$) and 40 µM TcdB-GTD. The ligand was titrated into the protein solution over 19 injections of 2 µL of 6 s with a 150 s equilibration period between injections. For binding measurements involving UDP and isofagomine and noeuromycin, 40 µM TcdB-GTD was incubated with 1 mM UDP in the buffer described above for 30 min in the sample cell prior to titrating isofagomine (250 µM) or noeuromycin (400 µM) that was prepared in the reaction buffer described above and 1 mM UDP. The resulting data were fit to a model of one distinct binding site. The first injection for each sample was excluded from data fitting. Titrations were run past the point of enzyme saturation to correct for heats of dilution. A separate titration without TcdB-GTD was run for each condition and used to subtract the background heat of dilution.

**Crystallization.** Co-crystallization of TcdB-GTD with UDP, isofagomine or noeuromycin used 10 mg/mL TcdB-GTD mixed with inhibitors (isofagomine or noeuromycin) and UDP in a 1:5:10 molar ratio and incubated for 2 h on ice. Crystallization experiments used sitting drop vapor diffusion at 22 °C. The TcdB-GTD:UDP:isofagomine or TcdB-GTD:UDP:noeuromycin complex was screened with the microlytic (MCSG1-4) and Hampton crystallization screens. Crystallization was performed using the CRYSTAL-GRYPHON crystallization robot (Art Robbins) in 96-well INTELLI plates (Art Robbins). The crystallization drop contained 0.5 µL TcdB-GTD-inhibitor mixture and 0.5 µL of well solution. The volume of the well solution was 70 µL. Diffracting crystals were obtained in two weeks. The complex of TcdB-GTD with UDP-isofagomine crystallized in 200 mM lithium sulfate and 20% (w/v) polyethylene glycol 3,350. TcdB-GTD with UDP and noeuromycin crystallized in 85 mM sodium citrate pH 5.6, 170 mM ammonium acetate, 25.5 % (w/v) PEG 4000 and 15 %(v/v) glycerol. The crystals of TcdB-GTD:UDP:isofagomine complex were cryoprotected with 20% ethylene glycol. Crystals were flash-frozen in liquid nitrogen for X-ray diffraction data collection.

**Diffraction data collection and processing**. Diffraction data from TcdB-GTD:UDP:isofagomine crystals were collected at the LRL-CAT beam line at 0.97931 Å wavelength (Argonne National Laboratory, Argonne, IL). The data for cdB-GTD:UDP:noeuromycin crystals were collected at NSLS-II Beamline at 0.97895 Å wavelength (Brookhaven National Laboratory, Upton NY) (Supplementary Table 1). The data were processed using the iMOSFLM and scaled by the AIMLESS program of the CCP4 suite (Supplementary Table 1)[62,63]. The data quality of crystal diffraction was analyzed by the SFCHECK and XTRIAGE programs[63,64]. Matthews coefficient (Vm) calculations were done to calculate monomers present in the asymmetric unit. The data collection and processing statistics are summarized in Table S1.

**Structure determination and refinement**. Crystal structures of TcdB-GTD:UDP:isofagomine and TcdB-GTD:UDP:noeuromycin were solved by molecular replacement using PHASER[65]. Chain-A of TcdB-GTD structure (PDB ID: 5UQM) was used as the initial phasing model. The model obtained from PHASER was manually adjusted and completed using the graphics program COOT[66]. Structure refinement was performed by REFMAC5 program, using standard protocols for the NCS refinement[67]. The final refinement statistics of the structure are summarized in Table S1.

**Structure analysis**. The crystal structure of apo TcdA-GTD (PDB ID: 4DMV), a structure in complex with UDP-glucose (PDB ID: 3SRZ) and the TcdB-GTD in complex with UDP-2-deoxy-2-fluoroglucose (PDB ID: 5UQN) were used for structural comparisons. Structural superimpositions used the SSM protocol of COOT. The geometry analyses of the final model used MolProbity[68]. The B-factor analysis of the structures used the BAVERAGE program of the CCP4 suite. Structural figures were produced by molecular graphics program PyMOL. For TcdB-GTD complexes, subunit-A was used for all the structural analyses and comparisons.

**Cell rounding assays**. Cell rounding assays were performed using Vero cells and IMR90 cells grown in EMEM medium with 10% FBS, HT-29 cells grown in McCoys 5 A medium with 10% FBS and CHO-K1 cells grown in F12-K medium with 10% FBS in 5% CO$_2$ at 37 °C. Cells were acquired from ATCC, and maintained with standard trypsin passaging protocols. For quantitation of cell rounding, Vero cells were seeded into 24-well plates at 50,000 cells/well. The following day, samples treated with isofagomine or noeuromycin (Serial dilution from 200 μM to 0.78 μM final concentration) were incubated for 20 min before addition of toxin. Cells were treated with 1 nM TcdA for 2 h or 1 pM TcdB for 1.5 h at 37 °C 5% CO$_2$ before imaging. Samples were imaged by light microscopy on a Zeiss Axioscope with 4 representative images taken for each sample. Rounded cells were counted for each image and expressed as percentage of total cells. Images were processed in FIJI/ImageJ, with manipulation limited to alterations of brightness and contrast.

**Western blot analysis of Rac1 glucosylation in IMR90 cells**. IMR90 cells were grown in EMEM with 10% FBS in 24-well plates inoculated at 50,000 cells/well. The next day isofagomine or noeuromycin was added to the wells at varying concentrations (12.5 μM, 25 μM, 50 μM and 100 μM) and cells were incubated at 37 °C, 5% CO$_2$ for 15 min. Next, TcdA (1 nM final) or TcdB (0.1 nM final) was added to each well and cells were incubated at 37 °C, 5% CO$_2$ for 6 h. The cells were lysed with mammalian cell lysis buffer (Abcam) according to manufacturer's instructions. Cell lysates (2 μg) were heated at 95 °C in XT-loading dye (Bio-rad) before being loaded on an SDS polyacrylamide gel. Following electrophoresis, samples were transferred to PVDF membrane using an iblot2 device (ThermoFisher), blocked with 1% milk in PBS 0.1% Tween 20 for 1 h at room temperature and probed with a 1/4000 dilution of anti-Rac1 (Mab102, Thermo Fisher) and 1/8000 of anti-GAPDH antibody (Abcam) for 1 h at room temperature. Following incubation with the primary antibody, the membrane was washed 3 times with PBS with 0.1% Tween20 and incubated with a 1/40,000 dilution of secondary antibody for 1 h. Following incubation with the secondary antibody, the membrane was washed 3 times in PBS 0.1% tween 20 before a final wash in PBS. To determine total Rac1 levels the same procedure was followed with the exceptions of blocking in 1% milk in PBS 0.1% Tween 20 overnight at 4 °C, incubation with 1/8000 dilution of anti-Rac1 (23A8, Millipore Sigma) for 45 min and incubation with a 1/80,000 dilution of secondary antibody (HRP Goat Anti-Mouse Ig, BD Biosciences) for 45 min. Chemiluminescent detection was carried out using Lumigen ECL reagent and images were taken using an ImageQuant LAS4000 mini (GE). Uncropped and unprocessed scans are provided in Supplementary Figs. 9, 10.

**FACS analysis of Annexin V staining of HT-29 cells**. HT-29 cells were grown in McCoys 5A media supplemented with 10% FBS, and seeded in 6-well plates at a density of 300,000 cells per well. The next day, isofagomine or noeuromycin (500 μM or 250 μM) was added and incubated for 30 min. Next, TcdA (1 nM) was added to each well and cells were incubated at 37 °C, 5% CO$_2$ for 24 h. Cells were harvested using 0.25% trypsin, filtered through 40 μm cell strainer caps into 5 mL polystyrene tubes and pelleted by centrifugation (1500 rpm, 5 min). Cells were stained for AnnexinV-FITC (Biolegend #640945) as per the manufacturer's instructions. Briefly, cells were washed twice with staining buffer, then resuspended in 100 μL binding buffer. AnnexinV (5 μL) was added, and samples were incubated in the dark for 15 min at room temperature. Binding buffer (400 μL) was added and samples were analyzed on an LSRII flow cytometer (Becton Dickson). Data was analyzed using FlowJo (TreeStar) and Prism (Graphpad) software.

**Reporting summary**. Further information on experimental design is available in the Nature Research Reporting Summary linked to this paper.

## Data availability

X-Ray crystallography data for structures of TcdB-GTD in complex with iminosugars and UDP were deposited into the PDB under accession codes 7LOU and 7LOV. Source data are provided with this paper.

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

## Acknowledgements

This work was supported by NIH research grants GM041916 and AI150971. The Albert Einstein Crystallographic Core X-Ray diffraction facility is supported by NIH Shared Instrumentation Grant S10 OD020068. Data collection also involved resources of the Advanced Photon Source; a U.S. Department of Energy (DOE) Office of Science User Facility operated for the DOE Office of Science by Argonne National Laboratory under Contract No. DE-AC02-06CH11357. Use of the Lilly Research Laboratories Collaborative Access Team (LRL-CAT) beamline at Sector 31 of the Advanced Photon Source was provided by Eli Lilly Company, which operates the facility. The Center for BioMolecular Structure (CBMS) is primarily supported by the National Institutes of Health, National Institute of General Medical Sciences (NIGMS) through a Center Core P30 Grant (P30GM133893), and by the DOE Office of Biological and Environmental Research (KP1607011). As part of NSLS-II, a national user facility at Brookhaven National Laboratory (BNL), work performed at the CBMS is supported in part by the U.S. Department of Energy, Office of Science, Office of Basic Energy Sciences Program under contract number and DE-SC0012704. K.S.P was supported by an American Australian Association Education Fellowship. We thank Dr. Teresa V Bowman and Margaret A Potts for helpful discussions on FACS analysis of TcdA treated samples. We also thank Drs. Agnidipta Ghosh and Tyler L. Grove for collecting TcdB-GTD UDP-noeuromycin complex diffraction data at BNL.

## Author contributions

A.S.P performed KIE analysis, cell rounding dose–response studies and western blot analysis. A.S.P and B.L.A performed enzyme inhibition assays and ITC binding experiments. R.K.H. conducted the structure determinations and characterizations. K.S.P performed Annexin V FACS analysis and microscopy. P.C.T performed synthesis of noeuromycin. V.L.S. designed and supervised the project. All authors contributed to writing the manuscript.

## Competing interests

The authors declare no competing interests.
