## [Peer Review File · Nature Communications]

Inhibition of Clostridium difficile TcdA and TcdB toxins with transition state analoguesREVIEWER COMMENTS

Reviewer #1 (Remarks to the Author):

This manuscript describes the characterization of two toxins from *C. difficile*, with respect to their inhibition by iminosugars that are deemed to be analogues of the enzymes' transition states. Following the multiple kinetic isotope effect approach that is a hallmark of the Schramm group, the authors established that the transition state of one of the toxins, TcdB-GTD, is dissociative (SN1-like), with significant carbocation character, not unlike several other glycoside-modifying enzymes studied by the group in the past. Accordingly, they tested several natural products containing an endocyclic nitrogen, which was expected to be cationic under the experimental conditions and therefore mimic the transition state. Two of these, isofagomine and noeuromycin, were found to be moderately effective inhibitors of both TcdB and TcdA enzymes but exhibited much stronger affinity (as low as 16 nM) in the presence of the product UDP. These inhibitors were effective in cell culture at preventing apoptosis that is otherwise induced by TcdA and TcdB, indicating that inhibition of these enzymes via transition state mimicry is a potentially viable means for treatment of *C. difficile* infections.

Overall, the experiments were conducted well and I support publication after the following concerns are addressed. These comments are given in order of appearance in the document.

1. Why were kinetic isotope effects only measured for TcdB? Considering that the inhibitors were ~20-fold more potent vs. TcdA, it is suggestive that the two transition states could be somewhat different (with the inhibitors better mimicking the TcdA transition state). Similar arguments have been made by the Schramm group in their studies on other glycoside-modifying enzymes, such as purine nucleoside phosphorylase. Measurement of isotope effects for both enzymes could make a much more compelling argument in support of the transition state mimicry professed to be responsible for the observed inhibition in this study.
2. On line 129 and elsewhere, square brackets are used incorrectly around isotope descriptors. The use of brackets is reserved for modifying a chemical name, to be placed immediately in front of the name (the hyphen should be removed in front of UDP, for example, at the end of line 127). These brackets are missing on line 599. Brackets should be removed in all other contexts, such as when describing a kinetic isotope effect. Thus, line 129 should be written as, "Therefore, 1³H, 2³H and 6³H kinetic isotope effects were measured in pairs with [6¹⁴C]UDP-glucose serving as the control." (note also that "kinetic isotope effects" was missing from the original text).
3. In the paragraph beginning on line 142, hyperconjugation is being described as originating from the 2³-hydrogen atom, but this is inaccurate. Hyperconjugation occurs by orbital mixing from an occupied orbital into an immediately adjacent unoccupied orbital. Thus, the wording should be changed to describe the hyperconjugation from the $\sigma(\text{C-H})$ orbital at C2³ to the $\sigma^*(\text{C-O})$ orbital from the anomeric carbon to the UDP leaving group. Later, on line 148, the authors mention "a lack of hyperconjugation to the [2³H]." This should be written as "a lack of hyperconjugation from the $\sigma(\text{C-H})$ orbital at C2³".
4. On line 145, reference 42 is used to support the statement that the C2³-H2³ bond is nearly perpendicular to the C1³-UDP bond, but ref. 42 does not relate to this concept. An appropriate reference relating the relevant dihedral angle to the magnitude of the 2³-3H KIE should be included. The authors may wish to consult Sunko et al. JACS 1977, 5000 and/or previous studies from the Schramm group.
5. On lines 150-151, the term "hexopyrano" is used, but to the best of my knowledge, this is not a real term. Pyrano is a prefix that relates to pyran rather than to the carbohydrate pyranose. Thus, use hexopyranose.
6. On line 156, a KIE of 1.047 is cited from ref. 38, but I found a value of 1.0425 in this reference. It is mentioned twice in that article.
7. On line 157, "a fully protonated transition states" lacks singular-plural agreement. In the sentence that follows, the authors interpret the 1³-18O KIE as being the result of extensive bond cleavage at the transition state, reduced by protonation of the leaving group. While this is one possibility, so is having partial bond cleavage with minimal protonation of the leaving group. A justification in support of the former should be made; otherwise (or additionally), both mechanistic extremes should be mentioned. Furthermore, for the synthesis of the 18O-labeled substrate, was the isotopic enrichment assessed

(e.g., by mass spec)? I understand that radioactive material may not be compatible with the spectrometer, but a “cold” reaction run in parallel would be a reasonable compromise. If such analysis was performed, that information should be included. If not, the authors need to mention that their ^{18}O KIE is a lower limit, and it is possible that the extent of bond cleavage is larger and/or of protonation is smaller at the transition state.

8. On line 171, gluconolactone is described as a polyhydroxic acid. This molecule is not an acid, though it becomes one upon hydrolysis of the ester.

9. On line 201, the wording is confusing. The iminosugar would be expected to bind to an enzyme-UDP complex, not to form one. Perhaps just deleting the word “form” would suffice.

10. The results of the binding experiments point to ordered product release, with glucose before UDP. Evidence in support of this can be found in the structural work of D’Urzo et al. FEBS Journal, 2012, 3085, which should be mentioned.

11. References 40 and 41 are repeated on lines 408-415.

12. The numbering in Table 1 should include double prime. Also, if $6''\text{-}^{14}\text{C}$ was used as the remote label, why isn’t its KIE 1 by definition?

Reviewer #2 (Remarks to the Author):

In this manuscript, Paparella et al. describe the identification and characterization of small molecule inhibitors of the two disease-causing toxins produced by *C. difficile*. The authors elegantly and convincingly used kinetic isotope effects to show that the glucosyltransferase domain (GTD) of these toxins form a glucocation-like transition state in the hydrolase reaction. Based on this, the authors tested potential transition state mimics and identified two, isofagomine and noeuromycin, that effectively inhibited both the TcdB-GTD and TcdA-GTD. The authors next used ITC to carefully show the mode of binding and describe the properties of the interaction of the two molecules. Next, the authors solved high-resolution structures of the two TS mimics with TcdB-GTD revealing the binding site and molecular interactions of the two compounds in the active site. Finally, the authors tested both compounds for their ability to inhibit TcdA and TcdB induced toxicity on mammalian cells - showing that these compounds are able to inhibit the cytotoxic effects of these compounds by inhibiting GTD-mediated glucosylation of Rac1, albeit at very high molar concentrations.

This work is of major importance both for the field of toxin biology and for developing therapeutics against these devastating proteins. I was very impressed with the depth of the analysis in this paper and the number of different techniques used to characterize these compounds. The only possible weakness of the paper was the rather modest inhibition of the two compounds on cells. Because authors did not perform a proper dose-titration on cells, it is unclear what the exact EC₅₀ of each compound is, but it appears to be approaching 100uM - potencies that are unlikely to have any clinical benefit at normal dose ranges, notwithstanding the remarkable PK of these molecules. Additional experiments as suggested below may help refine this and put these numbers into better context.

Major points:

- In Figure 4, the authors used 0.1 TcdB to test the efficacy of their compounds. These doses are relatively very high for TcdB, which normally causes cell-rounding at 1pM or lower after 3 hours. This point is important because at 0.1nM, TcdB begins to cause cellular toxicity that is INDEPENDENT of GTD activity. The so-called necrosis phenotype has been shown to not require enzyme activity per se, and so testing compounds against the GTD at this dose may be misleading.
- The authors should repeat these experiments at lower doses of TcdB, and also do a proper dose titration with compound to calculate the IC₅₀ or EC₅₀.
- Can the authors comment on the cell permeability of isofagomine and Noe? One wonders if this is the source of the large shift in potency from in vitro to cells. It would be nice for this to be tested formally, but if they cannot, a discussion on the topic would suffice.

Minor points:

- In the introduction, the authors need to add references in the second paragraph when they describe how TcdA/B get into cells (i.e., between references 12 and 13)
- The first section of the results could benefit from a re-write for brevity. The depth of detail is a little overwhelming for a journal like Nature Comms and unnecessary for the reader to understand the phenomena being described.
- Cell pictures: inclusion of size/scale bars would help the presentation of the figures
- For cell experiments, I recommend dose titration of the Isof and Noe compounds to estimate IC50 values
- Figure 4e: X-axis is mislabeled for noe (need to shift "250" and "500" to the right)
- In the discussion, can you expand on the pharmacokinetics of Isof and Noe, especially with regard to expected levels in the colon, where TcdB and TcdA are present?
- The authors may want to soften their language on the use of these compounds therapeutically. I think it would be more convincing of the authors instead proposed these findings as blue-prints for future compound development - either to use the x-ray structures to design new analogues, OR perhaps to pro-drug these compounds to increase their membrane permeability.

Reviewer #3 (Remarks to the Author):

CDI is a potentially life threatening disease. In this manuscript, Dr. Paparella and colleagues set out to look for transition state analogue inhibitors targeting the GTD of TcdB and TcdA. They found that isofagomine and noeuromycin were the most potent inhibitors that exhibited an uncompetitive inhibition, but in a UDP-dependent manner. However, the lead compounds were not evaluated systemically in vitro, and their cell-based assays were also problematic. Testing these compounds in a CDI animal disease model was completely missing. Furthermore, similar studies on TcdB/TcdA have been reported before. Therefore, this manuscript does not represent the type of advance that is suitable for Nature Communications. Some more specific comments are listed below.

1. Fig. 1 may be useful for a review paper, but not here.
2. Line 164: has glucose been tested?
3. For the ITC studies, 40 μM TcdB-GTD was incubated with 1 mM UDP, but how much isofagomine and noeuromycin was use? Was UDP included in the syringe buffer? Was buffer titration done? Why the N values were everywhere? For a K_d at about 67 nM as reported here, 40 μM TcdB-GTD for ITC was too much.
4. How to understand their results showing that UDP itself has a K_d around 49 μM , while isofagomine could reach 67 nM (with UDP)?
5. The K_i of isofagomine for TcdB-GTD was improved from about 1.4 μM (no UDP) to 0.29 μM (with UDP), not a big difference. This seems to contradict their ITC studies showing no binding of isof in the absence of UDP, but nM binding in the presence of UDP.
6. The crystal structures are nice. But what can we learn from their structures? Do their structures help to understand why some compounds are better than the other? Castanospermine was also reported as a transition state mimic inhibitor (Ref. 42). Why is it worse than the two reported here?
7. Statement in lines 258-260 is misleading, as the observed loop conformational change is unlikely to be caused by the inhibitors.
8. Cell rounding assay was done using a single dose of isof at 100 μM . Needs careful titration and quantification (not just showing 3 images). TcdB was used at 0.1 nM, which was quite high for cell rounding assay. Why was this dose chosen?
9. For glucosylation assay shown in Fig. 4d, how many repeats were done? Quantification?
10. It is alarming that only one single dose at 250 μM (isofagomine) or 500 μM (noeuromycin) were reported in Fig. 4e.
11. Why used three different cell lines for three assays: CHO-K1 for cell rounding, IMR90 for Rac1 glucosylation assay, and HT-29 for apoptosis? It is hard to align these results as different cell lines have different expression patterns of TcdA/B receptors.
12. In general, it is difficult to obtain a selective glycosyltransferase inhibitor only based on a donor

structure, which raises concerns about hitting off targets. There are bigger problems for the two compounds reported here, because they both need to rely on UDP. In a physiologically relevant setting, when UDP becomes available after UDP-glucose cleavage, glucose will be readily transferred to Rho proteins, and damage is done. So, how realistic is it for such compounds to be protective?

13. Please add the fragment ranges for the recombinant GTD and Rac1.

14. The figure legend for Fig. S7b is wrong.

Reviewer comments are *in italics*, responses are in blue.

Reviewer #1 summarized the review with the comment: “Overall, the experiments were conducted well and I support publication after the following concerns are addressed. These comments are given in order of appearance in the document.”

1. *Why were kinetic isotope effects only measured for TcdB? Considering that the inhibitors were ~20-fold more potent vs. TcdA, it is suggestive that the two transition states could be somewhat different (with the inhibitors better mimicking the TcdA transition state). Similar arguments have been made by the Schramm group in their studies on other glycoside-modifying enzymes, such as purine nucleoside phosphorylase. Measurement of isotope effects for both enzymes could make a much more compelling argument in support of the transition state mimicry professed to be responsible for the observed inhibition in this study.*

We thank the reviewer for raising an important point in the manuscript. KIEs on the glycohydrolase reaction of TcdA have been done and are included in the revision. We now summarize the important transition state features for the glycohydrolase reactions of both TcdA and TcdB. The expanded KIEs are presented in Table 1. We found that the TcdA GH transition state has a higher 1-³H KIE and a lower 1-¹⁴C KIE. The TcdA GH transition state has more glucocation character than TcdB. This explains why isofagomine and noeuromycin are better inhibitors of the TcdA enzyme. We now discuss this difference in the main text.

2. *On line 129 and elsewhere, square brackets are used incorrectly around isotope descriptors. The use of brackets is reserved for modifying a chemical name, to be placed immediately in front of the name (the hyphen should be removed in front of UDP, for example, at the end of line 127). These brackets are missing on line 599. Brackets should be removed in all other contexts, such as when describing a kinetic isotope effect. Thus, line 129 should be written as, “Therefore, 1³H, 2³H and 6³H kinetic isotope effects were measured in pairs with [6¹⁴C]UDP-glucose serving as the control.” (note also that “kinetic isotope effects” was missing from the original text).*

We thank the reviewer for noting this. We have made the corrections in the manuscript text.

3. *In the paragraph beginning on line 142, hyperconjugation is being described as originating from the 2³H-hydrogen atom, but this is inaccurate. Hyperconjugation occurs by orbital mixing from an occupied orbital into an immediately adjacent unoccupied orbital. Thus, the wording should be changed to describe the hyperconjugation from the $\sigma(\text{C-H})$ orbital at C2³ to the $\sigma^*(\text{C-O})$ orbital from the anomeric carbon to the UDP leaving group. Later, on line 148, the authors mention “a lack of hyperconjugation to the [2³H].” This should be written as “a lack of hyperconjugation from the $\sigma(\text{C-H})$ orbital at C2³.”*

We thank the reviewer for this clarification. We have edited the text accordingly to read:

The β -secondary $2''$ - ^3H KIE reports on the degree of hyperconjugation that occurs from the $\sigma(\text{C-H})$ orbital at $\text{C}2''$ to the $\sigma^*(\text{C-O})$ orbital from the anomeric carbon to the UDP leaving group^{35,36}. The $2''$ - ^3H KIE for TcdB-GTD was measured to be 1.014 ± 0.001 and 1.052 ± 0.005 for TcdA-GTD. The magnitude of both $2''$ - ^3H KIEs suggests that the $\text{C}2''\text{-H}2''$ bond is near-perpendicular to the $\text{C}1''\text{-UDP}$ bond at the transition state³⁶. Other N-glycohydrolases and glycosyltransferases with glucocation-like transition states express β -secondary $2''$ - ^3H KIEs >1.07 ^{35,37}. As the TcdB-GTD and TcdA-GTD $1''$ - ^{14}C and $1''$ - ^3H KIEs support a glucocationic transition state, the $2''$ - ^3H GH KIEs indicate an unusual transition state geometry with a lack of hyperconjugation from the $\sigma(\text{C-H})$ orbital at $\text{C}2''$.

4. On line 145, reference 42 is used to support the statement that the $\text{C}2''\text{-H}2''$ bond is nearly perpendicular to the $\text{C}1''\text{-UDP}$ bond, but ref. 42 does not relate to this concept. An appropriate reference relating the relevant dihedral angle to the magnitude of the $2''$ - ^3H KIE should be included. The authors may wish to consult Sunko et al. JACS 1977, 5000 and/or previous studies from the Schramm group.

We thank the reviewer. We have reassigned the correct reference.

5. On lines 150-151, the term "hexopyrano" is used, but to the best of my knowledge, this is not a real term. Pyrano is a prefix that relates to pyran rather than to the carbohydrate pyranose. Thus, use hexopyranose.

We thank the reviewer and now use hexopyranose.

6. On line 156, a KIE of 1.047 is cited from ref. 38, but I found a value of 1.0425 in this reference. It is mentioned twice in that article.

We thank the reviewer for finding this mis-reference. The correct reference, where the maximal leaving group KIE of 1.047 is reported for β -glucosidases, is now referenced. Rosenberg, S. & Kirsch J. F. Oxygen-18 leaving group kinetic isotope effects on the hydrolysis of nitrophenyl glycosides. 2. Lysozyme and β -glucosidase: acid and alkaline hydrolysis. *Biochemistry* **20**, 3196-3204, (1981).

7. On line 157, "a fully protonated transition states" lacks singular-plural agreement. In the sentence that follows, the authors interpret the $1''$ - ^{18}O KIE as being the result of extensive bond cleavage at the transition state, reduced by protonation of the leaving group. While this is one possibility, so is having partial bond cleavage with minimal protonation of the leaving group. A justification in support of the former should be made; otherwise (or additionally), both mechanistic extremes should be mentioned. Furthermore, for the synthesis of the ^{18}O -labeled substrate, was the isotopic enrichment assessed (e.g., by mass spec)? I understand that radioactive material may not be compatible with the spectrometer, but a "cold" reaction run in parallel would be a reasonable compromise. If such analysis was performed, that information should be included. If not, the authors need to mention that their ^{18}O KIE is a lower limit, and it is possible that the extent of bond cleavage is larger and/or of protonation is smaller at the transition state.

The text now includes both possibilities for the leaving group KIE. The text now reads, "Therefore, the $1''$ - ^{18}O KIE measured in this study could reflect partial cleavage of the

glycosidic bond. Alternatively, leaving group ^{18}O KIEs for acid-catalyzed C-O bond cleavage of sugar glucosides thought to proceed through a fully protonated transition states are reported to give KIEs of 1.023 – 1.026^{41,42}. The $1''\text{-}^{18}\text{O}$ KIE measured here could be interpreted as either partial bond cleavage with minimal protonation of the leaving group or extensive bond cleavage at the transition state with protonation of the leaving group oxygen^{35,37}."

For the synthesis of the 1- ^{18}O UDP-glucose substrate, the methods state "For the synthesis of 1- ^{18}O , 6- ^{14}C double labeled UDP-glucose, the isotope enrichment was assessed via mass spectrometry on a 'cold' reaction that was run in parallel. Mass analysis indicated ^{18}O substitution of >99%."

8. On line 171, gluconolactone is described as a polyhydroxic acid. This molecule is not an acid, though it becomes one upon hydrolysis of the ester.

We have edited the text to read, "Gluconolactone, a well described GH inhibitor...."

9. On line 201, the wording is confusing. The iminosugar would be expected to bind to an enzyme-UDP complex, not to form one. Perhaps just deleting the word "form" would suffice.

We edited the text to read, "The logical complex for iminosugar binding is to the enzyme-UDP complex."

10. The results of the binding experiments point to ordered product release, with glucose before UDP. Evidence in support of this can be found in the structural work of D'Urzo et al. FEBS Journal, 2012, 3085, which should be mentioned.

We thank the reviewer and have edited the text to include this information and updated reference.

11. References 40 and 41 are repeated on lines 408-415.

We thank the reviewer for finding this. We have edited the reference list accordingly.

12. The numbering in Table 1 should include double prime. Also, if 6''- ^{14}C was used as the remote label, why isn't its KIE 1 by definition?

Yes, that information is now added to Table 1 as a footnote.

Reviewer #2 summarized the work as:

"This work is of major importance both for the field of toxin biology and for developing therapeutics against these devastating proteins. I was very impressed with the depth of the analysis in this paper and the number of different techniques used to characterize these compounds. The only possible weakness of the paper was the rather modest inhibition of the two compounds on cells. Because authors did not perform a proper dose-titration on cells, it is unclear what the exact EC50 of each compound is, but it appears to be approaching 100uM - potencies that are unlikely to have any clinical benefit at normal dose ranges, notwithstanding the remarkable PK of these molecules. Additional experiments as suggested below may help refine this and put these numbers into better context."

Major points:

- In Figure 4, the authors used 0.1 nM TcdB to test the efficacy of their compounds. These doses are relatively very high for TcdB, which normally causes cell-rounding at 1pM or lower after 3 hours. This point is important because at 0.1nM, TcdB begins to cause cellular toxicity that is INDEPENDENT of GTD activity. The so-called necrosis phenotype has been shown to not require enzyme activity per se, and so testing compounds against the GTD at this dose may be misleading. The authors should repeat these experiments at lower doses of TcdB, and also do a proper dose titration with compound to calculate the IC50 or EC50.

We thank the reviewer for this comment. We added a proper dose titration with both isofagomine and noeuromycin to calculate the IC50 of cell rounding for each compound. For these experiments, we used Vero cells from the ATCC. Cells were pre-treated with either iminosugar for 20 minutes before addition of Toxin (1nM TcdA or 1 pM TcdB). These toxin levels are in the range recommended by the reviewer. Cells were incubated with toxin and inhibitors for 1.5 hr for TcdB and 2 hr for TcdA before images were taken. For each sample, 4 images for taken and the rounded cells were counted and expressed as a percentage of total cells. The IC50 for cell rounding ranges from 5.5 µM to 13.6 µM. We have included these results in Table 3 and in Figure 4 and S7 of the manuscript.

- Can the authors comment on the cell permeability of isofagomine and Noe? One wonders if this is the source of the large shift in potency from in vitro to cells. It would be nice for this to be tested formally, but if they cannot, a discussion on the topic would suffice.

Isofagomine reached maximal levels in the plasma, liver, spleen and brain of Sprague-Dawley rats within 1 hr after administration (ref Khanna et al, FEBS journal, 2010, vol 277, p 1618). Isofagomine was in phase 2 clinical trials for Gaucher's disease as an oral therapy, but did not improve clinical symptoms. These previous studies highlight tissue distribution and bioavailability of Isofagomine. We have included these additional references in the manuscript. This is the first study that reports noeuromycin efficacy in human cells. Noeuromycin protects against TcdA and TcdB in Fibroblast cells, CHO-K1 cells , HT-29 cells, Vero cells with IC50 values similar to Isofagomine. Thus, both have permeability, now discussed in the manuscript.

Minor points:

- In the introduction, the authors need to add references in the second paragraph when they describe how TcdA/B get into cells (i.e., between references 12 and 13)

The introduction cites three reviews on the mechanism of TcdA and TcdB action in references 12-14. All of them describe how Tcds get into cells, and the information is summarized in our introduction.

- The first section of the results could benefit from a re-write for brevity. The depth of detail is a little overwhelming for a journal like Nature Comms and unnecessary for the reader to understand the phenomena being described.

We have condensed the section by moving some of the text to the methods.

- Cell pictures: inclusion of size/scale bars would help the presentation of the figures

A scale bar has been added for the images presented in figure 4.

- For cell experiments, I recommend dose titration of the Isof and Noe compounds to estimate IC50 values

We have included a dose titration of Isofagomine and noeuromycin to calculate the IC50 values. These are presented in revised Table 3 and Figure S7.

- Figure 4e: X-axis is mislabeled for noe (need to shift "250" and "500" to the right)

We thank the reviewer for finding the label error. The image in Figure 4c is now corrected.

- In the discussion, can you expand on the pharmacokinetics of Isof and Noe, especially with regard to expected levels in the colon, where TcdB and TcdA are present?

A paragraph has been added to the discussion to discuss potential PK of Isof and Noe.

- The authors may want to soften their language on the use of these compounds therapeutically. I think it would be more convincing of the authors instead proposed these findings as blue-prints for future compound development - either to use the x-ray structures to design new analogues, OR perhaps to pro-drug these compounds to increase their membrane permeability.

These issues 'softened' in the discussion of potential utility of Isof and Noe.

Reviewer #3 summarized the opinion:

"CDI is a potentially life threatening disease. In this manuscript, Dr. Paparella and colleagues set out to look for transition state analogue inhibitors targeting the GTD of TcdB and TcdA. They found that isofagomine and noeuromycin were the most potent inhibitors that exhibited an uncompetitive inhibition, but in a UDP-dependent manner. However, the lead compounds were not evaluated systemically in vitro, and their cell-based assays were also problematic. Testing these compounds in a CDI animal disease model was completely missing. Furthermore, similar studies on TcdB/TcdA have been reported before. Therefore, this manuscript does not represent the type of advance that is suitable for Nature Communications. Some more specific comments are listed below."

1. Fig. 1 may be useful for a review paper, but not here.

We disagree with the reviewer's comments. Publication has the goal of reaching a broad journal audience. Figure 1 is useful for the vast majority of readers unfamiliar with mechanism of action of *C. difficile* toxins.

2. Line 164: has glucose been tested?

Glucose shows no significant inhibition. It has also been tested in reference 42 against TcdB-GTD with no significant inhibition.

3. For the ITC studies, 40 μM TcdB-GTD was incubated with 1 mM UDP, but how much isofagomine and noeuromycin was use? Was UDP included in the syringe buffer? Was buffer titration done? Why the N values were everywhere? For a K_d at about 67 nM as reported here, 40 μM TcdB-GTD for ITC was too much.

We have now edited the method text with better clarity. The methods now include the concentrations of noeuromycin, isofagomine and UDP included in the syringe buffer. Control buffer titrations were done for each condition (Isof/noe/UDP alone and Isof+UDP and noe+UDP) and subtracted from the experimental data to correct for heats of dilution. These are described in the methods.

4. How to understand their results showing that UDP itself has a K_d around 49 μM , while isofagomine could reach 67 nM (with UDP)?

The dissociation constant for $\text{Enz} + \text{UDP} \leftrightarrow \text{Enz-UDP}$ is 49 μM . The dissociation constant for $\text{Enz-UDP} + \text{Isof} \leftrightarrow \text{Enz-UDP-Isof}$ is 67 nM. This is explained carefully in the manuscript.

5. The K_i of isofagomine for TcdB-GTD was improved from about 1.4 μM (no UDP) to 0.29 μM (with UDP), not a big difference. This seems to contradict their ITC studies showing no binding of isof in the absence of UDP, but nM binding in the presence of UDP.

In steady-state kinetic assays UDP is formed as a product from UDP-glucose. Therefore the steady-state amount of enzyme-UDP is present at all times. As not all of the enzyme is in this form the K_i differs from K_d by ITC. These are standard kinetic principles. ITC experiments quantitated the binding show no binding to TcdB alone, proving that Isofagomine binds only to the enzyme-UDP complex.

6. The crystal structures are nice. But what can we learn from their structures? Do their structures help to understand why some compounds are better than the other? Castanospermine was also reported as a transition state mimic inhibitor (Ref. 42). Why is it worse than the two reported here?

Isofagomine and noeuromycin are glucocation mimics. The transition state is a glucocation shown by our experimental KIE experiments. We appreciate the reviewer pointing out the careless and incorrect use of the 'transition state' designation in the literature. Castanospermine is not a good glucocation mimic and is a poor inhibitor. The X-ray crystal structures demonstrate that isofagomine and noeuromycin form ion pair interactions between the endocyclic nitrogen atoms and the β -phosphate of UDP. They occupy the glucose space, demonstrating action as transition state analogs. The structures

will also serve as blueprints for the design of second generation TcdA/TcdB transition state analogue inhibitors.

7. *Statement in lines 258-260 is misleading, as the observed loop conformational change is unlikely to be caused by the inhibitors.*

We thank the reviewer for their comment. We agree, the text has been edited to read:

Apo TcdB-GTD structures have loops (Glu449 to Asp461 and Gln510 to Asp523) in open conformations (Figures 3c and S6). These loops are closed in the UDP inhibitor-bound structures which is consistent with previously published structures of the Tcd toxins in complex with UDP-glucose and UDP^{47,49-51}.

8. *Cell rounding assay was done using a single dose of isof at 100 μ M. Needs careful titration and quantification (not just showing 3 images). TcdB was used at 0.1 nM, which was quite high for cell rounding assay. Why was this dose chosen?*

We determined cell protection by titration of isofagomine and noeuromycin in the presence of TcdA and TcdB to quantitate cell rounding. We used Vero cells treated with either 1 nM TcdA or 1 pM TcdB. The IC₅₀ for each compound is now reported in the new experiments reported here in Table 3 and in Figures 4a and S7.

9. *For glycosylation assay shown in Fig. 4d, how many repeats were done? Quantification?*

For Figure 4d, each western blot was done at least twice. Thus has now been included in the figure/legend/methods section. We did not quantify the band corresponding to unglucosylated Rac1.

10. *It is alarming that only one single dose at 250 μ M (isofagomine) or 500 μ M (noeuromycin) were reported in Fig. 4e.*

The experimental methods/figure legend have been clarified to show that the FACS analysis of AnnexinV, was performed with both isofagomine and noeuromycin, each using 250 μ M and 500 μ M.

11. *Why used three different cell lines for three assays: CHO-K1 for cell rounding, IMR90 for Rac1 glycosylation assay, and HT-29 for apoptosis? It is hard to align these results as different cell lines have different expression patterns of TcdA/B receptors.*

All cell lines have been used in published studies reporting Tcd toxin action. We show protection by our inhibitors in all cell lines. We have utilized several human cell lines in this study to demonstrate the robustness/validity of our findings. We can confirm that the observed effect is not due to the specifics of a single immortalized cell line, but rather consistently occurring in multiple systems.

12. *In general, it is difficult to obtain a selective glycosyltransferase inhibitor only based on a donor structure, which raises concerns about hitting off targets. There are bigger problems for*

the two compounds reported here, because they both need to rely on UDP. In a physiologically relevant setting, when UDP becomes available after UDP-glucose cleavage, glucose will be readily transferred to Rho proteins, and damage is done. So, how realistic is it for such compounds to be protective?

The use of multiple cell lines demonstrating low micromolar protection demonstrates sufficient UDP is present in each system to permit protective action against the toxins. That there was no cell toxicity at high isofagomine and noeuromycin treatment concentrations supports our conclusion of low off-target effects.

13. *Please add the fragment ranges for the recombinant GTD and Rac1.*

The methods now include the fragment ranges for TcdB-GTD and Rac1:
TcdB-GTD amino acids: 1-546
TcdA-GTD amino acids 1-542
Rac1 amino acids 1-192

14. *The figure legend for Fig. S7b is wrong.*

Figure S7 has been replaced with the dose response curves of Isofagomine/noeuromycin protection of Vero cell rounding.

REVIEWER COMMENTS

Reviewer #1 (Remarks to the Author):

In the most recent version of their manuscript, Paparella et al. have satisfactorily addressed all but one of the concerns raised in their initial submission. In my review of the initial manuscript, I asked, "For the synthesis of the ^{18}O -labeled substrate, was the isotopic enrichment assessed (e.g., by mass spec)? I understand that radioactive material may not be compatible with the spectrometer, but a "cold" reaction run in parallel would be a reasonable compromise. If such analysis was performed, that information should be included." The authors' response to this was, "For the synthesis of 1- ^{18}O , 6- ^{14}C double labeled UDP-glucose, the isotope enrichment was assessed via mass spectrometry on a 'cold' reaction that was run in parallel. Isotope enrichment was assessed to be $>99\%$." I should note that this text is found in the "marked up" version of the revised manuscript, but the final sentence is absent from the main PDF (see line 663).

I have concern about the claim that the ^{18}O enrichment was found to be $>99\%$. Writing this as a limit suggests that perhaps no ^{16}O isotopologue could be detected in their mass spectrum. Most spectrometers would be capable of detecting even 1% of the lighter isotopologue, so this claim is surprising. The isotope enrichment in the ^{18}O -water sold by Cambridge Isotope Labs was not provided by the authors (it should be stated) but is sold as 97% according to the company's website. Ignoring the small amount of isotope dilution caused by dissolving the glucose (0.25 mmol) in the labeled water (~ 14 mmol), the theoretical limit of ^{18}O enrichment in the exchanged product should be 97%. Including the isotope dilution, this is closer to 95%. Because of this discrepancy, I would like for the authors to supply a sample mass spectrum in the SI. Further, the methods section should include a description of the mass spectrometer and how the isotope ratio was calculated (assuming there is a peak for the light isotopologue).

Reviewer #2 (Remarks to the Author):

The authors have addressed all of my concerns. The paper is much clearer now.

Reviewer comments are *in italics*, responses are in blue.

Reviewer #1 (Remarks to the Author):

In the most recent version of their manuscript, Paparella et al. have satisfactorily addressed all but one of the concerns raised in their initial submission. In my review of the initial manuscript, I asked, "For the synthesis of the ^{18}O -labeled substrate, was the isotopic enrichment assessed (e.g., by mass spec)? I understand that radioactive material may not be compatible with the spectrometer, but a "cold" reaction run in parallel would be a reasonable compromise. If such analysis was performed, that information should be included." The authors' response to this was, "For the synthesis of 1- ^{18}O , 6- ^{14}C double labeled UDP-glucose, the isotope enrichment was assessed via mass spectrometry on a 'cold' reaction that was run in parallel. Isotope enrichment was assessed to be >99%." I should note that this text is found in the "marked up" version of the revised manuscript, but the final sentence is absent from the main PDF (see line 663).

I have concern about the claim that the ^{18}O enrichment was found to be >99%. Writing this as a limit suggests that perhaps no ^{16}O isotopologue could be detected in their mass spectrum. Most spectrometers would be capable of detecting even 1% of the lighter isotopologue, so this claim is surprising. The isotope enrichment in the ^{18}O -water sold by Cambridge Isotope Labs was not provided by the authors (it should be stated) but is sold as 97% according to the company's website. Ignoring the small amount of isotope dilution caused by dissolving the glucose (0.25 mmol) in the labeled water (~14 mmol), the theoretical limit of ^{18}O enrichment in the exchanged product should be 97%. Including the isotope dilution, this is closer to 95%. Because of this discrepancy, I would like for the authors to supply a sample mass spectrum in the SI. Further, the methods section should include a description of the mass spectrometer and how the isotope ratio was calculated (assuming there is a peak for the light isotopologue).

We thank the reviewer for their comments. We have edited the text in the materials and methods to read:

For the synthesis of 1- ^{18}O , 6- ^{14}C double labeled UDP-glucose, the isotope enrichment was assessed via mass spectrometry on a 'cold' reaction that was run in parallel. Isotope enrichment was assessed to be at least 95%. A sample mass spectrum is presented in figure S12. Mass spectra were acquired on a Shimadzu LCMS-2010EV spectrometer.

In addition, we have also prepared a figure (Figure S12) of a sample mass spectrum of a 'cold reaction' that was prepared in parallel to 1- ^{18}O , 6- ^{14}C double labeled UDP-glucose.